# Hamiltonian $Q$-Learning: Leveraging Importance-sampling for Data Efficient RL

## Abstract

Model-free reinforcement learning (RL), in particular $Q$-learning is widely used to learn optimal policies for a variety of planning and control problems. However, when the underlying state-transition dynamics are stochastic and high-dimensional, $Q$-learning requires a large amount of data and incurs a prohibitively high computational cost. In this paper, we introduce Hamiltonian $Q$-Learning, a data efficient modification of the $Q$-learning approach, which adopts an importance-sampling based technique for computing the $Q$ function. To exploit stochastic structure of the state-transition dynamics, we employ Hamiltonian Monte Carlo to update $Q$ function estimates by approximating the expected future rewards using $Q$ values associated with a subset of next states. Further, to exploit the latent low-rank structure of the dynamic system, Hamiltonian $Q$-Learning uses a matrix completion algorithm to reconstruct the updated $Q$ function from $Q$ value updates over a much smaller subset of state-action pairs. By providing an efficient way to apply $Q$-learning in stochastic, high-dimensional problems, the proposed approach broadens the scope of RL algorithms for real-world applications, including classical control tasks and environmental monitoring.

## 1 Introduction

In recent years, reinforcement learning (Sutton & Barto, 2018) have achieved remarkable success with sequential decision making tasks especially in complex, uncertain environments. RL algorithms have been widely applied to a variety of real world problems, such as resource allocation (Mao et al., 2016), chemical process optimization (Zhou et al., 2017), automatic control (Duan et al., 2016), and robotics (Kober et al., 2013). Existing RL techniques often offer satisfactory performance only when it is allowed to explore the environment long enough and generating a large amount of data in the process (Mnih et al., 2015; Kamthe & Deisenroth, 2018; Yang et al., 2020a). This can be prohibitively expensive and thereby limits the use of RL for complex decision support problems.

$Q$-Learning (Watkins, 1989; Watkins & Dayan, 1992) is a model-free RL framework that captures the salient features of sequential decision making, where an agent, after observing current state of the environment, chooses an action and receives a reward. The action chosen by the agent is based on a policy defined by the state-action value function, also called the $Q$ function. Performance of such policies strongly depends on the accessibility of a sufficiently large data set covering the space spanned by the state-action pairs. In particular, for high-dimensional problems, existing model-free RL methods using random sampling techniques leads to poor performance and high computational cost. To overcome this challenge, in this paper we propose an intelligent sampling technique that exploits the inherent structures of the underlying space related to the dynamics of the system.

It has been observed that formulating planning and control tasks in a variety of dynamical systems such as video games (Atari games), classical control problems (simple pendulum, cart pole and double integrator) and adaptive sampling (ocean sampling, environmental monitoring) as $Q$-Learning problems leads to low-rank structures in the $Q$ matrix (Ong, 2015; Yang et al., 2020b; Shah et al., 2020). Since these systems naturally consist of a large number of states, efficient exploitation of low rank structure of the $Q$ matrix can potentially lead to significant reduction in computational complexity and improved performance. However, when the state space is high-dimensional and further, the state transition is probabilistic, high computational complexity associated with calculating the expected $Q$ values of next states renders existing $Q$-Learning methods impractical.

A potential solution for this problem lies in approximating the expectation of $Q$ values of next states with the sample mean of $Q$ values over a subset of next states. A natural way to select a subset of next states is by drawing IID samples from the transition probability distribution. However, this straight forward approach becomes challenging when the state transition probability distribution is high-dimensional and is known only up to a constant. We address this problem by using Hamilton Monte Carlo (HMC) to sample next states; HMC draws samples by integrating a Hamiltonian dynamics governed by the transition probability (Neal et al., 2011). We improve the data efficiency further by using matrix completion methods to exploit the low rank structure of a $Q$ matrix.

RELATED WORK

**Data efficient Reinforcement Learning:** The last decade has witnessed a growing interest in improving data efficiency in RL methods by exploiting emergent global structures from underlying system dynamics. Deisenroth & Rasmussen (2011); Pan & Theodorou (2014); Kamthe & Deisenroth (2018); Buckman et al. (2018) have proposed model-based RL methods that improve data efficiency by explicitly incorporating prior knowledge about state transition dynamics of the underlying system. Dearden et al. (1998); Koppel et al. (2018); Jeong et al. (2017) propose Baysean methods to approximate the $Q$ function. Ong (2015) and Yang et al. (2020b) consider a model-free RL approach that exploit structures of state-action value function. The work by Ong (2015) decomposes the $Q$ matrix into a low-rank and sparse matrix model and uses matrix completion methods (Candes & Plan, 2010; Wen et al., 2012; Chen & Chi, 2018) to improve data efficiency. A more recent work by Yang et al. (2020b) has shown that incorporation of low rank matrix completion methods for recovering $Q$ matrix from a small subset of $Q$ values can improve learning of optimal policies. At each time step the agent chooses a subset of state-action pairs and update their $Q$ value according to the Bellman optimally equation that considers a discounted average between reward and expectation of the $Q$ values of next states. Shah et al. (2020) extends this work by proposing a novel matrix estimation method and providing theoretical guarantees for the convergence to a $\epsilon$-optimal $Q$ function. On the other hand, entropy regularization (Ahmed et al., 2019; Yang et al., 2019; Smirnova & Dohmatob, 2020), by penalizing excessive randomness in the conditional distribution of actions for a given state, provides an alternative means to implicitly exploit the underlying low-dimensional structure of the value function. Lee et al. (2019) proposes an approach that samples a whole episode and then updates values in a recursive, backward manner.

CONTRIBUTION

The main contribution of this work is three-fold. *First*, we introduce a modified $Q$-learning framework, called Hamiltonian $Q$-learning, which uses HMC sampling for efficient computation of $Q$ values. This innovation, by proposing to sample $Q$ values from the region with the dominant contribution to the expectation of discounted reward, provides a data-efficient approach for using $Q$-learning in real-world problems with high-dimensional state space and probabilistic state transition. Furthermore, integration of this sampling approach with matrix-completion enables us to update $Q$ values for only a small subset of state-action pairs and thereafter reconstruct the complete $Q$ matrix. *Second*, we provide theoretical guarantees that the error between the optimal $Q$ function and the $Q$ function obtained by updating $Q$ values using HMC sampling can be made arbitrarily small. This result also holds when only a handful of $Q$ values are updated using HMC and the rest are estimated using matrix completion. Further, we provide theoretical guarantee that the sampling complexity of our algorithm matches the mini-max sampling complexity proposed by Tsybakov (2008). *Finally*, we demonstrate the effectiveness of Hamiltonian $Q$-learning by applying it to a cart-pole stabilization problem and an adaptive ocean sampling problem. Our results also indicate that our proposed approach becomes more effective with increase in state space dimension.

## 2 PRELIMINARY CONCEPTS

In this section, we provide a brief background on $Q$-Learning, HMC sampling and matrix completion, as well as introduce the mathematical notations. In this paper, $|\mathcal{Z}|$ denotes the cardinality of a set $\mathcal{Z}$. Moreover, $\mathbb{R}$ represent the real line and $A^T$ denotes the transpose of matrix $A$.

### 2.1 $Q$-LEARNING

Markov Decision Process (MDP) is a mathematical formulation that captures salient features of sequential decision making (Bertsekas, 1995). In particular, a *finite MDP* is defined by the tuple

$(\mathcal{S}, \mathcal{A}, \mathbb{P}, r, \gamma)$, where $\mathcal{S}$ is the finite set of system states, $\mathcal{A}$ is the finite set of actions, $\mathbb{P} : \mathcal{S} \times \mathcal{A} \times \mathcal{S} \to [0, 1]$ is the transition probability kernel, $r : \mathcal{S} \times \mathcal{A} \to \mathbb{R}$ is a bounded reward function, and $\gamma \in [0, 1)$ is a discounting factor. Without loss of generality, states $s \in \mathcal{S}$ and actions $a \in \mathcal{A}$ can be assumed to be $\mathcal{D}_s$-dimensional and $\mathcal{D}_a$-dimensional real vectors, respectively. Moreover, by letting $s^i$ denote the $i$th element of a state vector, we define the range of state space in terms of the following intervals $[d_i^-, d_i^+]$ such that $s^i \in [d_i^-, d_i^+] \ \forall i \in \{1, \ldots, \mathcal{D}_s\}$. At each time $t \in \{1, \ldots, T\}$ over the decision making horizon, an agent observes the state of the environment $s_t \in \mathcal{S}$ and takes an action $a_t$ according to some policy $\pi$ which maximizes the discounted cumulative reward. Once this action has been executed, the agent receives a reward $r(s_t, a_t)$ from the environment and the state of the environment changes to $s_{t+1}$ according to the transition probability kernel $\mathbb{P}(\cdot|s_t, a_t)$. The $Q$ function, which represents the expected discounted reward for taking a specific action at the current time and following the policy thereafter, is defined as a mapping from the space of state-action pairs to the real line, i.e. $Q : \mathcal{S} \times \mathcal{A} \to \mathbb{R}$. Then, by letting $Q^t$ represent the $Q$ matrix at time $t$, i.e. the tabulation of $Q$ function over all possible state-action pairs associated with the finite MDP, we can express the $Q$ value iteration over time steps as

$$Q^{t+1}(s_t, a_t) = \sum_{s \in \mathcal{S}} \mathbb{P}(s|s_t, a_t) \left( r(s_t, a_t) + \gamma \max_a Q^t(s, a) \right). \tag{1}$$

Under this update rule, the $Q$ function converges to its unique optimal value $Q^*$ (Melo, 2001). But computing the sum (1) over all possible next states is computationally expensive in certain problems; in these cases taking the summation over a subset of the next states provides an efficient alternative for updating the $Q$ values.

## 2.2 HAMILTONIAN MONTE CARLO

Hamiltonian Monte Carlo is a sampling approach for drawing samples from probability distributions known up to a constant. It offers faster convergence than Markov Chain Monte Carlo (MCMC) sampling (Neal et al., 2011; Betancourt; Betancourt et al., 2017; Neklyudov et al., 2020). To draw samples from a smooth target distribution $\mathcal{P}(s)$, which is defined on the Euclidean space and assumed to be known up to a constant, HMC extends the target distribution to a joint distribution over the target variable $s$ (viewed as position within the HMC context) and an auxiliary variable $v$ (viewed as momentum within the HMC context). We define the Hamiltonian of the system as

$$H(s, v) = -\log \mathcal{P}(s, v) = -\log \mathcal{P}(s) - \log \mathcal{P}(v|s) = U(s) + K(v, s),$$

where $U(s) \triangleq -\log \mathcal{P}(s)$ and $K(v, s) \triangleq -\log \mathcal{P}(v|s) = \frac{1}{2} v^T M^{-1} v$ represent the potential and kinetic energy, respectively, and $M$ is a suitable choice of the mass matrix.

HMC sampling method consists of the following *three* steps $-$ (i) a new momentum variable $v$ is drawn from a fixed probability distribution, typically a multivariate Gaussian; (ii) then a new proposal $(s', v')$ is obtained by generating a trajectory that starts from $(s, v)$ and obeys Hamiltonian dynamics, i.e. $\dot{s} = \frac{\partial H}{\partial v}, \dot{v} = -\frac{\partial H}{\partial s}$; and (iii) finally this new proposal is accepted with probability $\min\{1, \exp(H(s, v) - H(s', -v'))\}$ following the Metropolis–Hastings acceptance/rejection rule.

## 2.3 LOW-RANK STRUCTURE IN $Q$-LEARNING AND MATRIX COMPLETION

Prior work (Johns & Mahadevan, 2007; Geist & Pietquin, 2013; Ong, 2015; Shah et al., 2020) on value function approximation based approaches for RL has implicitly assumed that the state-action value functions are low-dimensional and used various basis functions to represent them, e.g. CMAC, radial basis function, etc. This can be attributed to the fact that the underlying state transition and reward function are often endowed with some structure. More recently, Yang et al. (2020b) provide empirical guarantees that the $Q$-matrices for benchmark Atari games and classical control tasks exhibit low-rank structure.

Therefore, using matrix completion techniques (Xu et al., 2013; Chen & Chi, 2018) to recover $Q \in \mathbb{R}^{|\mathcal{S}| \times |\mathcal{A}|}$ from a small number of observed $Q$ values constitutes a viable approach towards improving data efficiency. As low-rank matrix structures can be recovered by constraining the nuclear norm, the $Q$ matrix can be reconstructed from its observed values ($\hat{Q}$) by solving

$$Q = \underset{\widetilde{Q} \in \mathbb{R}^{|\mathcal{S}| \times |\mathcal{A}|}}{\arg \min} \|\widetilde{Q}\|_* \quad \text{subject to} \quad \mathcal{J}_\Omega(\widetilde{Q}) = \mathcal{J}_\Omega(\hat{Q}), \tag{2}$$

where $\| \cdot \|_*$ denotes the nuclear norm (i.e., the sum of its singular values), $\Omega$ is the observed set of elements, and $\mathcal{J}_\Omega$ is the observation operator, i.e. $\mathcal{J}_\Omega(x) = x$ if $x \in \Omega$ and zero otherwise.

# 3 HAMILTONIAN $Q$-LEARNING

A large class of real world sequential decision making problems - for example, board/video games, control of robots' movement, and portfolio optimization - involves high-dimensional state spaces and often has large number of distinct states along each individual dimension. As using a $Q$-Learning based approach to train RL-agents for these problems typically requires tens to hundreds of millions of samples (Mnih et al., 2015; Silver et al., 2017), there is a strong need for data efficient algorithms for $Q$-Learning. In addition, state transition in such systems is often probabilistic in nature; even when the underlying dynamics of the system is inherently deterministic; presence of external disturbances and parameter variations/uncertainties lead to probabilistic state transitions.

Learning an optimal $Q^*$ function through value iteration methods requires updating $Q$ values of state-action pairs using a sum of the reward and a discounted expectation of $Q$ values associated with next states. In this work, we assume the reward to be a deterministic function of state-action pairs. However, when the reward is stochastic, these results can be extended by replacing the reward with its expectation. Subsequently, we can express (1) as

$$Q^{t+1}(s_t, a_t) = r(s_t, a_t) + \gamma \mathbb{E}\left(\max_a Q^t(s, a)\right), \tag{3}$$

where $\mathbb{E}$ denotes the expectation over the discrete probability measure $\mathbb{P}$. When the underlying state space is high-dimensional and has large number of states, obtaining a more accurate estimate of the expectation is computationally very expensive. The complexity increases quadratically with the number of states and linearly with number of actions, rendering the existing algorithms impractical.

In this work, we propose a solution to this issue by introducing an importance-sampling based method to approximate the aforementioned expectation from a sample mean of $Q$ values over a subset of next states. A natural way to sample a subset from the set of all possible next states is to draw identically and independently distributed (IID) samples from the transition probability distribution $\mathbb{P}(\cdot|s_t, a_t)$. However, when the transition probability distribution is high-dimensional and known only up to a constant, drawing IID samples incurs a very high computation cost.

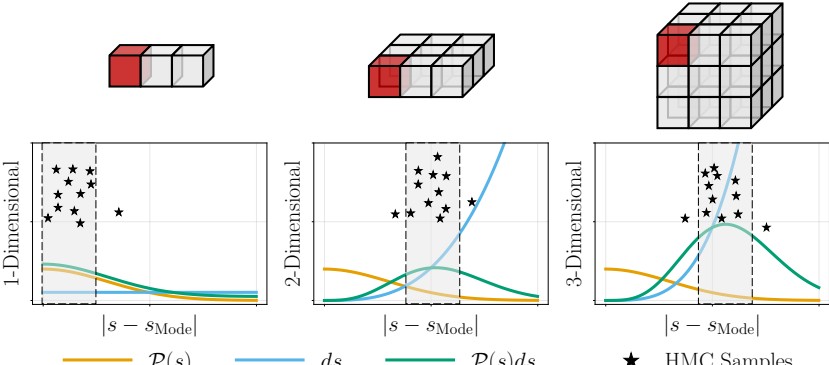

Figure 1: The first row illustrates that, as the dimension of the space increases, the relative volume inside a partition compared to the volume outside of the partition decreases. When the dimension increases from 1 through 3, the relative volume of red partition decreases as 1/3, 1/9 and 1/27, respectively. The second row illustrates that the HMC samples are concentrated in the region that maximizes probability mass. Here $\mathcal{P}(s)$, $s_{\text{Mode}}$ and $ds$ represent probability density, mode of the distribution and volume, respectively.

## 3.1 DATA EFFICIENCY THROUGH HMC SAMPLING

A number of importance-sampling methods (Liu, 1996; Betancourt) have been developed for estimating the expectation of a function by drawing samples from the region with the dominant contribution to the expectation. HMC is one such importance-sampling method that draws samples from the typical set, i.e., the region that maximizes probability mass, which provides the dominated contribution to the expectation. As shown in the second row of Figure 1, most of the samples in a limited pool of HMC samples indeed concentrate around the region with high probability mass. Since the decay in $Q$ function is significantly smaller compared to the typical exponential or power law decays in transition probability function, HMC provides a better approximation for the expectation of the $Q$ value of the next states (Yang et al., 2020b; Shah et al., 2020). Then by letting $\mathcal{H}_t$ denote the set of

HMC samples drawn at time step $t$, we update the $Q$ values as:

$$Q^{t+1}(s_t, a_t) = r(s_t, a_t) + \frac{\gamma}{|\mathcal{H}_t|} \sum_{s \in \mathcal{H}_t} \max_a Q^t(s, a). \tag{4}$$

**HMC for a smooth truncated target distribution:** Recall that region of states is a subset of a Euclidean space given as $s \in [d_1^-, d_1^+] \times \ldots \times [d_{\mathcal{D}_s}^-, d_{\mathcal{D}_s}^+] \subset \mathbb{R}^{\mathcal{D}_s}$. Thus the main challenge to using HMC sampling is to define a smooth continuous target distribution $\mathcal{P}(s|s_t, a_t)$ which is defined on $\mathbb{R}^{\mathcal{D}_s}$ with a sharp decay at the boundary of the region of states (Yi & Doshi-Velez, 2017; Chevallier et al., 2018). In this work, we generate the target distribution by first defining the transition probability kernel from the conditional probability distribution defined on $\mathbb{R}^{\mathcal{D}_s}$ and then multiplying it with a smooth cut-off function.

We first consider a probability distribution $\mathcal{P}(\cdot|s_t, a_t) : \mathbb{R}^{\mathcal{D}_s} \to \mathbb{R}$ such that the following holds

$$\mathbb{P}(s|s_t, a_t) \propto \int_{s-\varepsilon}^{s+\varepsilon} \mathcal{P}(s|s_t, a_t) ds \tag{5}$$

for some arbitrarily small $\varepsilon > 0$. Then the target distribution can be defined as

$$\mathcal{P}(s|s_t, a_t) = \mathcal{P}(s|s_t, a_t) \prod_{i=1}^{\mathcal{D}_s} \frac{1}{1 + \exp(-\kappa(d_i^+ - s^i))} \frac{1}{1 + \exp(-\kappa(s^i - d_i^-))}. \tag{6}$$

Note that there exists a large $\kappa > 0$ such that if $s \in [d_1^-, d_1^+] \times \ldots \times [d_{\mathcal{D}_s}^-, d_{\mathcal{D}_s}^+]$ then $\mathcal{P}(s|s_t, a_t) \propto \mathbb{P}(s|s_t, a_t)$ and $\mathcal{P}(s|s_t, a_t) \approx 0$ otherwise. Let $\mu(s_t, a_t), \Sigma(s_t, a_t)$ be the mean and covariance of the transition probability kernel. In this paper we consider transition probability kernels of the form

$$\mathbb{P}(s|s_t, a_t) \propto \exp\left(-\frac{1}{2}(s - \mu(s_t, a_t))^T \Sigma^{-1}(s_t, a_t)(s - \mu(s_t, a_t))\right). \tag{7}$$

Then from (5) the corresponding mapping can be given as a multivariate Gaussian $\mathcal{P}(s|s_t, a_t) = \mathcal{N}(\mu(s_t, a_t), \Sigma(s_t, a_t))$. Thus from (6) it follows that the target distribution is

$$\mathcal{P}(s|s_t, a_t) = \mathcal{N}(\mu(s_t, a_t), \Sigma(s_t, a_t)) \prod_{i=1}^{\mathcal{D}_s} \frac{1}{1 + \exp(-\kappa(d_i^+ - s^i))} \frac{1}{1 + \exp(-\kappa(s^i - d_i^-))} \tag{8}$$

**Choice of potential energy, kinetic energy and mass matrix:** Recall that the target distribution $\mathcal{P}(s|s_t, a_t)$ is defined over the Euclidean space $\mathbb{R}^{\mathcal{D}_s}$. For brevity of notation we drop the explicit dependence on $(s_t, a_t)$ and denote the target distribution as $\mathcal{P}(s)$. As explained in Section 2.2 we choose the potential energy $U(s) = -\log(\mathcal{P}(s))$. We consider an Euclidean metric $\mathcal{M}$ that induces the distance between $\tilde{s}, \bar{s}$ as $d(\tilde{s}, \bar{s}) = (\tilde{s} - \bar{s})^T \mathcal{M}(\tilde{s} - \bar{s})$. Then we define $\mathcal{M}_s \in \mathbb{R}^{\mathcal{D}_s \times \mathcal{D}_s}$ as a diagonal scaling matrix and $\mathcal{M}_r \in \mathbb{R}^{\mathcal{D}_s \times \mathcal{D}_s}$ as a rotation matrix in dimension $\mathcal{D}_s$. With this we can define $M$ as $M = \mathcal{M}_r \mathcal{M}_s \mathcal{M} \mathcal{M}_s^T \mathcal{M}_r^T$. Thus, any metric $M$ that defines an Euclidean structure on the target variable space induces an inverse structure on the momentum variable space as $d(\tilde{v}, \bar{v}) = (\tilde{v} - \bar{v})^T M^{-1}(\tilde{v} - \bar{v})$. This generates a natural family of multivariate Guassian distributions such that $\mathcal{P}(v|s) = \mathcal{N}(0, M)$ leading to the kinetic energy $K(v, s) = -\log \mathcal{P}(v|s) = \frac{1}{2} v^T M^{-1} v$ where $M^{-1}$ is the covariance of the target distribution.

### 3.2 $Q$-LEARNING WITH HMC AND MATRIX COMPLETION

In this work we consider problems with a high-dimensional state space and large number of distinct states along individual dimensions. Although these problems admit a large $Q$ matrix, we can exploit low rank structure of the $Q$ matrix to further improve the data efficiency.

At each time step $t$ we randomly sample a subset $\Omega_t$ of state-action pairs (each state-action pair is sampled independently with probability $p$) and update the $Q$ function for state-action pairs in $\Omega_t$. Let $\widehat{Q}^{t+1}$ be the updated $Q$ matrix at time $t$. Then from (4) we have

$$\widehat{Q}^{t+1}(s_t, a_t) = r(s_t, a_t) + \frac{\gamma}{|\mathcal{H}_t|} \sum_{s \in \mathcal{H}_t} \max_a Q^t(s, a), \qquad \forall (s_t, a_t) \in \Omega_t. \tag{9}$$

Then we recover the complete matrix $Q^{t+1}$ by using the method given in (2). Thus we have

$$Q^{t+1} = \underset{\widetilde{Q}^{t+1} \in \mathbb{R}^{|\mathcal{S}| \times |\mathcal{A}|}}{\arg\min} \|\widetilde{Q}^{t+1}\|_* \quad \text{subject to} \quad \mathcal{J}_{\Omega_t}\left(\widetilde{Q}^{t+1}\right) = \mathcal{J}_{\Omega_t}\left(\widehat{Q}^{t+1}\right). \tag{10}$$

---

**Algorithm 1** Hamiltonian $Q$-Learning

---

**Inputs:** Discount factor $\gamma$; Range of state space; Time horizon $T$;
**Initialization:** Randomly initialize $Q^0$
**for** $t = 1$ **to** $T$ **do**
    Step 1: Randomly sample a subset of state-action pairs $\Omega_t$
    Step 2: **HMC sampling phase** - Sample a set of next states $\mathcal{H}_t$ according to the target distribution defined in (6)
    Step 3: **Update phase** - For all $(s_t, a_t) \in \Omega_t$
    $\widehat{Q}^{t+1}(s_t, a_t) = r(s_t, a_t) + \frac{\gamma}{|\mathcal{H}_t|} \sum_{s \in \mathcal{H}_t} \max_a Q^t(s, a)$
    Step 4: **Matrix Completion phase**
    $Q^{t+1} = \arg\min_{\widetilde{Q}^{t+1} \in \mathbb{R}^{|\mathcal{S}| \times |\mathcal{A}|}} \|\widetilde{Q}^{t+1}\|_* \quad \text{subject to} \quad \eth_{\Omega_t}\left(\widetilde{Q}^{t+1}\right) = \eth_{\Omega_t}\left(\widehat{Q}^{t+1}\right)$
**end for**

---

Similar to the approach used by Yang et al. (2020b), we approximate the rank of the $Q$ matrix as the minimum number of singular values that are needed to capture 99% of its nuclear norm.

### 3.3 CONVERGENCE, BOUNDEDNESS AND SAMPLING COMPLEXITY

In this section we provide the main theoretical results of this paper. First, we formally introduce the following *regularity assumptions*:
(**A1**) The state space $\mathcal{S} \subseteq \mathbb{R}^{\mathcal{D}_s}$ and the action space $\mathcal{A} \subseteq \mathbb{R}^{\mathcal{D}_a}$ are compact subsets.
(**A2**) The reward function is bounded, i.e., $r(s, a) \in [R_{\min}, R_{\max}]$ for all $(s, a) \in \mathcal{S} \times \mathcal{A}$.
(**A3**) The optimal value function $Q^*$ is $C$-Lipschitz, i.e.

$$\left| Q^*(s, a) - Q^*(s', a') \right| \leq C\left( \|s - s'\|_F + \|a - a'\|_F \right)$$

where $\|\cdot\|_F$ is the Frobenius norm (which is same as the Euclidean norm for vectors).

We provide theoretical guarantees that Hamiltonian $Q$-Learning converges to an $\epsilon$-optimal $Q$ function with $\widetilde{O}\left(\frac{1}{\epsilon^{\mathcal{D}_s + \mathcal{D}_a + 2}}\right)$ number of samples. This matches the mini-max lower bound $\Omega\left(\frac{1}{\epsilon^{\mathcal{D}_s + \mathcal{D}_a + 2}}\right)$ proposed in Tsybakov (2008). First we define a family of $\epsilon$-optimal $Q$ functions as follows.

**Definition 1** ($\epsilon$-**optimal** $Q$ **functions**). *Let $Q^*$ be the unique fixed point of the Bellman optimality equation given as $(\mathcal{T}Q)(s', a') = \sum_{s \in \mathcal{S}} \mathbb{P}(s|s', a') \left( r(s', a') + \gamma \max_a Q(s, a) \right) \ \forall (s', a') \in \mathcal{S} \times \mathcal{A}$ where $\mathcal{T}$ denotes the Bellman operator. Then, under update rule (3), the $Q$ function almost surely converges to the optimal $Q^*$. We define $\epsilon$-optimal $Q$ functions as the family of functions $\mathbf{Q}_\epsilon$ such that $\|Q' - Q^*\|_\infty \leq \epsilon$ whenever $Q' \in \mathbf{Q}_\epsilon$.*

As $\|Q' - Q^*\|_\infty = \max_{(s,a) \in \mathcal{S} \times \mathcal{A}} \|Q'(s, a) - Q^*(s, a)\|$, any $\epsilon$-optimal $Q$ function is element wise $\epsilon$-optimal. Our next result shows that under HMC sampling rule given in Step 3 of the Hamiltonian $Q$-Learning algorithm (Algorithm 1), the $Q$ function converges to the family of $\epsilon$-optimal $Q$ functions.

**Theorem 1** (**Convergence of** $Q$ **function under HMC**). *Let $\mathcal{T}$ be an optimality operator under HMC given as $(\mathcal{T}Q)(s', a') = r(s', a') + \frac{\gamma}{|\mathcal{H}|} \sum_{s \in \mathcal{H}} \max_a Q(s, a), \ \forall (s', a') \in \mathcal{S} \times \mathcal{A}$, where $\mathcal{H}$ is a subset of next states sampled using HMC from the target distribution given in (6). Then, under update rule (4) and for any given $\epsilon \geq 0$, there exists $n_{\mathcal{H}}, t' > 0$ such that $\|Q^t - Q^*\|_\infty \leq \epsilon \ \forall t \geq t'$.*

Refer Appendix A.1 for proof of this theorem. The next theorem shows that the $Q$ matrix estimated via a suitable matrix completion technique lies in the $\epsilon$-neighborhood of the corresponding $Q$ function obtained via exhaustive sampling.

**Theorem 2** (**Bounded Error under HMC with Matrix Completion**). *Let $Q_{\mathcal{E}}^{t+1}(s_t, a_t) = r(s_t, a_t) + \gamma \sum_{s \in \mathcal{S}} \mathbb{P}(s|s_t, a_t) \max_a Q_{\mathcal{E}}^t(s, a), \forall (s_t, a_t) \in \mathcal{S} \times \mathcal{A}$ be the update rule under exhaustive sampling, and $Q^t$ be the $Q$ function updated according to Hamiltonian $Q$-Learning (9)-(10). Then, for any given $\tilde{\epsilon} \geq 0$, there exists $n_{\mathcal{H}} = \min_\tau |\mathcal{H}_\tau|, t' > 0$, such that $\|Q^t - Q_{\mathcal{E}}^t\|_\infty \leq \tilde{\epsilon} \ \forall t \geq t'$.*

Please refer Appendix A.2 for proof of this theorem. Finally we provide guarantees on sampling complexity of Hamiltonian $Q$-Learning algorithm.

**Theorem 3.** (**Sampling complexity of Hamiltonian** $Q$-**Learning**) *Let $\mathcal{D}_s, \mathcal{D}_a$ be the dimension of state space and action space, respectively. Consider the Hamiltonian $Q$-Learning algorithm presented in Algorithm 1. Then, under a suitable matrix completion method, the $Q$ function convergea to the family of $\epsilon$-optimal $Q$ functions with $\widetilde{O}\left(\epsilon^{-(\mathcal{D}_s + \mathcal{D}_a + 2)}\right)$ number of samples.*

Proof of Theorem 3 is given in Appendix B.

## 4 EXPERIMENTS

### 4.1 EMPIRICAL EVALUATION FOR CART-POLE

**Experimental setup:** By letting $\theta, \dot{\theta}$ denote the angle and angular velocity of the pole and $x, \dot{x}$ denote the position and velocity of the cart, the 4-dimensional state vector for the cart-pole system can be defined as $s = (\theta, \dot{\theta}, x, \dot{x})$. After defining the range of state space as $\theta \in [-\pi/2, \pi/2]$, $\dot{\theta} \in [-3.0, 3.0]$, $x \in [-2.4, 2.4]$ and $\dot{x} \in [-3.5, 3.5]$, we define the range of the scalar action as $a \in [-10, 10]$. Then each state space dimension is discretized into 5 distinct values and the action space into 10 distinct values. This leads to a $Q$ matrix of size $625 \times 10$. To capture parameter uncertainties and external disturbances, we assume that the probabilistic state transition is governed by a multivariate Gaussian with zero mean and covariance $\Sigma = \text{diag}[0.143, 0.990, 0.635, 1.346]$. To drive the pole to an upright position, we define the reward function as $r(s, a) = \cos^4(15\theta)$ (Yang et al., 2020b). After initializing the $Q$ matrix using randomly chosen values from $[0, 1]$, we sample state-action pairs independently with probability $p = 0.5$ at each iteration. Additional experimental details and results are provided in Appendix C.

**Results:** As it is difficult to visualize a heat map for a 4-dimensional state space, we show results for the first two dimensions $\theta, \dot{\theta}$ with fixed $x, \dot{x}$. The color in each cell of the heat maps shown in Figures 2(a), 2(b) and 2(c) indicates the value of optimal action associated with that state. These figures illustrate that the policy heat map for Hamiltonian $Q$-Learning is closer to the policy heat map for $Q$-Learning with exhaustive sampling. The two curves in Figure 2(d), that show the Frobenius norm of the difference between the learned $Q$ function and the optimal $Q^*$, illustrate that Hamiltonian $Q$-Learning achieves better convergence than $Q$-Learning with IID sampling. We also show that the sampling efficiency of any $Q$-Learning algorithm can be significantly improved by incorporating Hamiltonian $Q$-Learning. We illustrate this by incorporating Hamiltonian $Q$-Learning with vanilla $Q$-Learning, DQN, Dueling DQN and DDPG. Figure 3 shows how Frobenius norm of the error between $Q$ function and the optimal $Q^*$ varies with increase in the number of samples. Red solid curves correspond to the case with exhaustive sampling and black dotted curves correspond to the case with Hamiltonian $Q$-Learning. These results illustrate that Hamiltonian $Q$-Learning converges to an $\epsilon$ optimal $Q$ function with significantly smaller number of samples than exhaustive sampling.

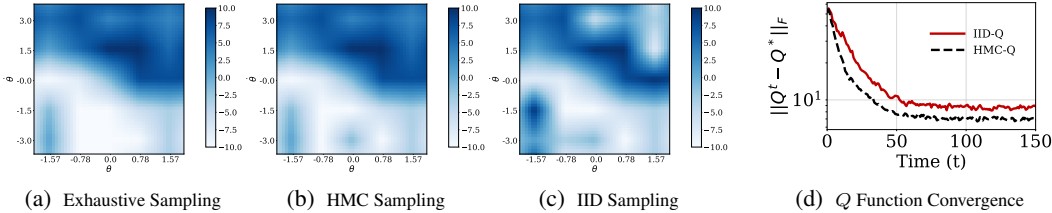

(a) Exhaustive Sampling    (b) HMC Sampling    (c) IID Sampling    (d) $Q$ Function Convergence

Figure 2: Figure 2(a), 2(b) and 2(c) show policy heat maps for $Q$-Learning with exhaustive sampling, Hamiltonian $Q$-Learning and $Q$-Learning with IID sampling, respectively. Figure 2(d) provides a comparison for convergence of $Q$ function with Hamiltonian $Q$-Learning and $Q$-Learning with IID sampling.

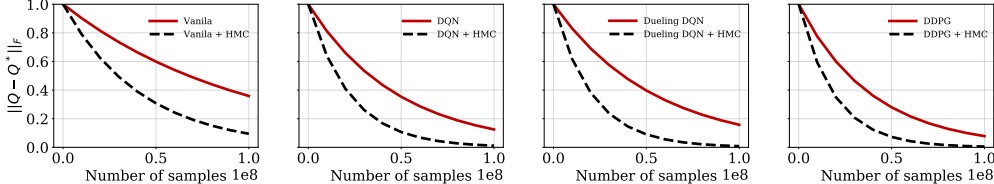

Figure 3: Mean square error vs number of samples of $Q$ function with exhaustive sampling and Hamiltonian $Q$-Learning for vanila $Q$-Learning, DQN, Dueling DQN and DDPG.

### 4.2 EMPIRICAL EVALUATION FOR ACROBOT (I.E., DOUBLE PENDULUM)

**Experimental setup:** By letting $\theta_1, \dot{\theta}_1, \theta_2, \dot{\theta}_2$ denote the angle of the first pole, angular velocity of the first pole, angle of the second pole and angular velocity of the second pole, respectively, the 4-dimensional state vector for the acrobot can be defined as $s = (\theta_1, \dot{\theta}_1, \theta_2, \dot{\theta}_2)$. After defining the

range of state space as $\theta_1 \in [-\pi, \pi]$, $\dot\theta_1 \in [-3.0, 3.0]$, $\theta_2 \in [-\pi, \pi]$ and $\dot\theta_2 \in [-3.0, 3.0]$, we define the range of the scalar action as $a \in [-10, 10]$. Then each state space dimension is discretized into 5 distinct values and the action space into 10 distinct values. This leads to a $Q$ matrix of size $625 \times 10$. Furthermore, we assume that the probabilistic state transition is governed by a multivariate Gaussian with zero mean and covariance $\Sigma = \text{diag}[0.143, 0.990, 0.635, 1.346]$. Following Sutton & Barto (2018), we define an appropriate reward function for stabilizing the acrobot to the upright position. After initializing the $Q$ matrix using randomly chosen values from $[0, 1]$, we sample state-action pairs independently with probability $p = 0.5$ at each iteration.

**Results:**   Figure 4 illustrates how Frobenius norm of the error between $Q$ function and the optimal $Q^*$ varies with the number of samples. Red solid curves correspond to the case with exhaustive sampling and black dotted curves correspond to the case with Hamiltonian $Q$-Learning. These results show that for the same level of error Hamiltonain $Q$-Learning requires a significantly smaller number of samples compared to exhaustive sampling.

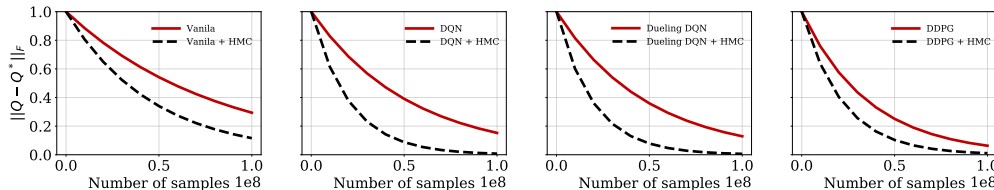

Figure 4: Mean square error vs number of samples of $Q$ function with exhaustive sampling and Hamiltonian $Q$-Learning for vanila $Q$-Learning, DQN, Dueling DQN and DDPG.

### 4.3   APPLICATION TO OCEAN SAMPLING

Ocean sampling plays a major role in a variety of science and engineering problems, ranging from modeling marine ecosystems to predicting global climate. Here, we consider the problem of using an under water glider to obtain measurements of a scalar field (e.g., temperature, salinity or concentration of a certain zooplankton) and illustrate how the use of Hamiltonian $Q$-Learning in planning the glider trajectory can lead to measurements that minimize the uncertainty associated with the field.

**States, actions and state transition:**   By assuming that the glider's motion is restricted to an horizontal plane (Refael & Degani, 2019), we let $x$, $y$ and $\theta$ denote its center of mass position and heading angle, respectively. Then we can define the 6-dimensional state vector for this system as $s = (x, y, \dot x, \dot y, \theta, \dot \theta)$ and the action $a$ as a scalar control input to the glider. Also, to accommodate dynamic perturbations due to the ocean current, other external disturbances and parameter uncertainties, we assume that the probabilistic state transition is governed by a multivariate Gaussian.

**Reward:**   As ocean fields often exhibit temporal and spatial correlations (Leonard et al., 2007), this work focuses on spatially correlated scalar fields. Following the approach of Leonard et al. (2007), we define ocean statistic correlation between two positions $\mathbf{q} = (x, y)$ and $\mathbf{q}' = (x', y')$ as $B(\mathbf{q}, \mathbf{q}') = \exp(-\|\mathbf{q} - \mathbf{q}'\|^2 / \sigma^2)$, where $\sigma$ is the spatial decorrelation scale. The goal of the task is to take measurements that reduce the uncertainty associated with the field. Now we assume that the glider takes $N$ measurements at positions $\{\mathbf{q}_1, \dots, \mathbf{q}_N\}$. Then covariance of the collected data set can be given by a $N \times N$ matrix $\mathcal{W}$ such that its $i$th row and the $j$th column element is: $\mathcal{W}_{ij} = \eta \delta_{ij} + B(\mathbf{q}_i, \mathbf{q}_j)$, where $\delta_{ij}$ is the Dirac delta and $\eta$ is the variance of the uniform and uncorrelated measurement noise. Then using objective analysis data assimilation scheme (Kalnay, 2003; Bennett, 2005), the total reduction of uncertainty of the field after the measurements at positions $\mathcal{Q} = \{\mathbf{q}_1, \dots, \mathbf{q}_N\}$ can be expressed as

$$\mathcal{U} = \sum_{\mathbf{q} \in \mathcal{Q}} \sum_{i,j=1}^{N} B(\mathbf{q}, \mathbf{q}_i) \mathcal{W}_{ij}^{-1} B(\mathbf{q}_j, \mathbf{q}), \tag{11}$$

By substituting the formulas from (Kalnay, 2003; Bennett, 2005) into (11), this formulation can be generalized to Gaussian measurement noise.

Recall that the task objective is to guide the glider to take measurements at multiple locations/positions which maximize the reduction in uncertainty associated with the scalar field. Therefore the reward assigned to each state-action pair $(s, a)$ is designed to reflect the amount of uncertainty that can

be reduced by taking a measurement at the position corresponding to the state and at the positions that correspond to the set of maximally probable next states, i.e., $\arg\max_{s'} \mathbb{P}(s'|s, a)$. Then, by letting $\mathcal{Z}_s = \{s\} \cup \{\cup_{a \in \mathcal{A}} \{\arg\max_{s'} \mathbb{P}(s'|s, a)\}\}$ denote the set of current state $s$ and the maximally probable next states for all possible actions, the reward function associated with reducing uncertainty can be given as

$$r_u(s, a) = \sum_{\mathbf{q} \in \mathcal{Q}} \sum_{i,j \in \mathcal{Z}_s} B(\mathbf{q}, \mathbf{q}_i) \mathcal{W}_{ij}^{-1} B(\mathbf{q}_j, \mathbf{q}).$$

Without loss of generality, we assume that the glider is deployed from $\mathbf{q} = (0, 0)$ and retrieving the glider incurs a cost depending on its position. To promote trajectories that do not incur a high cost for glider retrieval, we define the following reward function

$$r_c(s, a) = -\mathbf{q}^T \mathcal{C} \mathbf{q}$$

where $\mathcal{C} = \mathcal{C}^T \geq 0$. Then we introduce the total reward that aims to reduce uncertainty of the scalar field while penalizing the movements away from the origin, and define it as

$$r(s, a) = r_u(s, a) + r_c(s, a) = -\lambda \mathbf{q}^T \mathcal{C} \mathbf{q} + \sum_{\mathbf{q} \in \mathcal{Q}} \sum_{i,j \in \mathcal{Z}_s} B(\mathbf{q}, \mathbf{q}_i) \mathcal{W}_{ij}^{-1} B(\mathbf{q}_j, \mathbf{q}),$$

where $\lambda > 0$ is a trade-off parameter that maintains a balance between these two objectives.

**Experimental setup** We define the range of state and action space as $x, y \in [-10, 10]$, $\dot{x}, \dot{y} \in [-25, 25]$, $\theta \in [-\pi, \pi]$, $\dot{\theta} \in [-3, 3]$, and $a \in [-1, 1]$, respectively and then discretizing each state dimension into 5 distinct values and the action space into 5 distinct values, we have a $Q$ matrix of size $15625 \times 5$. Also, we assume that the state transition kernel is given by a multivariate Gaussian with zero mean and covariance $\Sigma = \text{diag}[11.111, 69.444, 11.111, 69.444, 0.143, 0.990]$. After initializing the $Q$ matrix using randomly chosen values from $[0, 1]$, we sample state-action pairs independently with probability $p = 0.5$ at each iteration. Also, we assume $\sigma = 2.5$, $\lambda = 0.1$, $\mathcal{C} = \text{diag}[1, 0]$. Additional experimental details and results are provided in Appendix D.

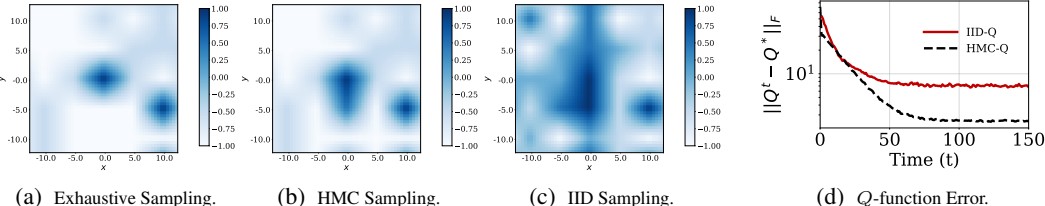

(a) Exhaustive Sampling.     (b) HMC Sampling.     (c) IID Sampling.     (d) $Q$-function Error.

Figure 5: Figure 5(a), 5(b) and 5(c) show policy heat maps for $Q$-Learning with exhaustive sampling, Hamiltonian $Q$-Learning and $Q$-Learning with IID sampling respectively. Figure 5(d) provides a comparison for convergence of $Q$ function with Hamiltonian $Q$-Learning and $Q$-Learning with IID sampling.

**Results** Figures 5(a), 5(b) and 5(c) show the policy heat map over first two dimensions $x, y$ with fixed $\dot{x}, \dot{y}, \theta$ and $\dot{\theta}$. The color of each cell indicates the value of optimal action associated with that state. These figures illustrate that the difference between policy heat maps associated with Hamiltonian $Q$-Learning and $Q$-Learning with exhaustive sampling is smaller than the difference between policy heat maps associated with $Q$-Learning with IID sampling and $Q$-Learning with exhaustive sampling. The two curves in Figure 5(d), that show the Frobenius norm of the difference between the learned $Q$ function and the optimal $Q^*$, illustrate that Hamiltonian $Q$-Learning achieves better convergence than $Q$-Learning with IID sampling. A comparison between results of the ocean sampling problem and the cart-pole stabilization problem indicates that Hamiltonian $Q$-Learning provides increasingly better performance with increase in state space dimension.

## 5 DISCUSSION AND CONCLUSION

Here we have introduced *Hamiltonian Q-Learning* which utilizes HMC sampling with matrix completion methods to improve data efficiency. We show, both theoretically and empirically, that the proposed approach can learn very accurate estimates of the optimal $Q$ function with much fewer data points. We also demonstrate that Hamiltonian $Q$-Learning performs significantly better than $Q$-Learning with IID sampling when the underlying state space dimension is large. By building upon this aspect, future works will investigate how importance-sampling based methods can improve data-efficiency in multi-agent Q-learning with agents coupled through both action and reward.

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

## A    CONVERGENCE AND BOUNDEDNESS RESULTS

We proceed to prove theorem by stating convergence properties for HMC as follows. In the initial sampling stage, starting from the initial position Markov chain converges towards to the typical set. In the next stage Markov chain quickly traverse the typical set and improves the estimate by removing the bias. In the last stage Markov chain refine the exploration of typical the typical set provide improved estimates. The number of samples taken during the last stage is referred as effective sample size.

### A.1    PROOF OF THEOREM 1

**Theorem 1.** *Let $\mathcal{T}$ be an optimality operator under HMC given as $(\mathcal{T}Q)(s', a') = r(s', a') + \frac{\gamma}{|\mathcal{H}|} \sum_{s \in \mathcal{H}} \max_a Q(s, a),\ \forall (s', a') \in \mathcal{S} \times \mathcal{A}$, where $\mathcal{H}$ is a subset of next states sampled using HMC from the target distribution given in (6). Then, under update rule (4) and for any given $\epsilon \geq 0$, there exists $n_{\mathcal{H}}, t' > 0$ such that $\|Q^t - Q^*\|_\infty \leq \epsilon\ \forall t \geq t'$.*

**Proof of Theorem 1.** Let $\bar{Q}^t(s, a) = \frac{1}{n_{\mathcal{H}}} \max_a Q^t(s, a), \forall (s, a) \in \mathcal{S} \times \mathcal{A}$. Here we consider $n_{\mathcal{H}}$ to be the effective number of samples. Let $\mathbb{E}_{\mathcal{P}} Q^t, \mathbf{Var}_{\mathcal{P}} Q^t$ be the expectation and covariance of $Q^t$ with respect to the target distribution. From Central Limit Theorem for HMC we have

$$\bar{Q}^t \sim \mathcal{N}\left(\mathbb{E}_{\mathcal{P}} Q^t, \sqrt{\frac{\mathbf{Var}_{\mathcal{P}} Q^t}{n_{\mathcal{H}}}}\right).$$

Since $Q$ function does not decay fast we provide a proof for the case where $Q^t$ is $C$-Lipschitz. From Theorem 6.5 in (Holmes et al., 2014) we have that, there exists a $c_0 > 0$ such that

$$||\bar{Q}^t - \mathbb{E}_{\mathcal{P}} Q^t|| \leq c_0. \tag{12}$$

Recall that Bellman optimality operator $\mathcal{T}$ is a contraction mapping. Thus from triangle inequality we have

$$\left\|\mathcal{T}Q_1 - \mathcal{T}Q_2\right\|_\infty \leq \max_{s',a'}\left\|r(s',a') + \frac{\gamma}{|\mathcal{H}_1|}\sum_{s\in\mathcal{S}}\max_a Q_1(s,a)\right.$$

$$\left. -r(s',a') - \frac{\gamma}{|\mathcal{H}_2|}\sum_{s\in\mathcal{S}}\max_a Q_2(s,a)\right\|$$

$$\leq \max_{s',a'}\left\|\frac{\gamma}{|\mathcal{H}_1|}\sum_{s\in\mathcal{S}}\max_a Q_1(s,a) - \frac{\gamma}{|\mathcal{H}_2|}\sum_{s\in\mathcal{S}}\max_a Q_2(s,a)\right\|$$

Let $|\mathcal{H}_1| = |\mathcal{H}_2| = n_\mathcal{H}$. Then using triangle inequality we have

$$\left\|\mathcal{T}Q_1 - \mathcal{T}Q_2\right\|_\infty \leq \max_{s',a'}\gamma\left[\left\|\bar{Q}_1 - \mathbb{E}_\mathcal{P}Q_1\right\| + \left\|\bar{Q}_2 - \mathbb{E}_\mathcal{P}Q_2\right\|\right] + \max_{s',a'}\gamma\left\|\mathbb{E}_\mathcal{P}Q_1 - \mathbb{E}_\mathcal{P}Q_2\right\|$$

Since $Q$ function almost surely converge under exhaustive sampling we have

$$\max_{s',a'}\gamma\left\|\mathbb{E}_\mathcal{P}Q_1 - \mathbb{E}_\mathcal{P}Q_2\right\| \leq \gamma\left\|Q_1 - Q_2\right\|_\infty \tag{13}$$

From equation 12 and equation 13 we have after $t$ time steps

$$\left\|\mathcal{T}Q_1 - \mathcal{T}Q_2\right\|_\infty \leq 2c_0 + \gamma\left\|Q_1 - Q_2\right\|_\infty$$

Let $R_{max}$ and $R_{min}$ be the maximum and minimum reward values. Then we have that

$$\left\|Q_1 - Q_2\right\|_\infty \leq \frac{\gamma}{1-\gamma}R_{max} - R_{min}.$$

Thus for any $\epsilon \geq$ by choosing a $\gamma$ such there exists a $t'$ such that $\forall t \geq t'$

$$\|Q^t - Q^*\|_\infty \leq \epsilon$$

This concludes the proof of Theorem 1. $\square$

## A.2 PROOF OF THEOREM 2

**Theorem 2.** *Let $Q_\mathcal{E}^{t+1}(s_t,a_t) = r(s_t,a_t) + \gamma\sum_{s\in\mathcal{S}}\mathbb{P}(s|s_t,a_t)\max_a Q_\mathcal{E}^t(s,a), \forall(s_t,a_t)\in\mathcal{S}\times\mathcal{A}$ be the update rule under exhaustive sampling, and $Q^t$ be the Q function updated according to Hamiltonian Q-Learning, i.e. by (9)-(10). Then, for any given $\tilde{\epsilon} \geq 0$, there exists $n_\mathcal{H}, t' > 0$, such that $\|Q^t - Q_\mathcal{E}^t\|_\infty \leq \tilde{\epsilon} \forall t \geq t'$.*

**Proof of Theorem 2.** Note that at each time step we attempt to recover the matrix $Q_\mathcal{E}^t$, i.e., $Q$ function time time $t$ under exhaustive sampling though a matrix completion method starting from $\widehat{Q}^t$, which is the $Q$ updated function at time $t$ using Hamiltonian $Q$-Learning. From Theorem 4 in (Chen & Chi, 2018) we have that $\forall t \geq t'$ there exists some constant $\delta > 0$ such that when the updated $Q$ function a $\widehat{Q}^t$ satisfy

$$\left\|\widehat{Q}^t - Q_\mathcal{E}^t\right\|_\infty \leq c$$

where $c$ is some positive constant then reconstructed (completed) matrix $Q^t$ satiesfies

$$\left\|Q^t - Q_\mathcal{E}^t\right\|_\infty \leq \delta\left|\widehat{Q}^t - Q_\mathcal{E}^t\right\|_\infty \tag{14}$$

for some $\delta > 0$. This implies that when the initial matrix used for matrix completion is sufficiently close to the matrix we are trying to recover matrix completion iterations converge to a global optimum. From the result of Theorem 1 we have for any given $\epsilon \geq 0$, there exists $n_\mathcal{H}, t' > 0$ such that $\forall t \geq t'$

$$\left\|\widehat{Q}^t - Q^*\right\| \leq \epsilon \tag{15}$$

Recall that under the update equation $Q_\mathcal{E}^{t+1}(s_t,a_t) = r(s_t,a_t) + \gamma\sum_{s\in\mathcal{S}}\max_a Q_\mathcal{E}^t(s,a), \forall(s_t,a_t)\in\mathcal{S}\times\mathcal{A}$ we have that $Q_\mathcal{E}$ almost surely converge to the optimal $Q^*$. Thus there exists a $t^\dagger$ such that $\forall t \geq t^\dagger$

$$\left\|Q_\mathcal{E}^t - Q^*\right\| \leq \epsilon$$

Let $t^\ddagger = \max\{t^\dagger, t'\}$. Then from triangle inequality we have that

$$\left\|\widehat{Q}^t - Q_\mathcal{E}^t\right\| \leq \left\|\widehat{Q}^t - Q^*\right\| + \left\|Q_\mathcal{E}^t - Q^*\right\| \leq 2\epsilon.$$

Thus from equation 14 we have that

$$\left\|Q^t - Q_{\mathcal{E}}^t\right\|_\infty \leq 2\delta\epsilon$$

This concludes the proof of Theorem 2. $\square$

# B  SAMPLING COMPLEXITY

In this section we provide theoretical results on sampling complexity of Hamiltonian $Q$-Learning. For brevity of notation we define $\mathcal{M}Q(s) = \max_a Q(s, a)$. Note that we have the following regularity conditions on the MDP studied in this paper.

**Regularity Conditions**

1. Spaces $\mathcal{S}$ and $\mathcal{A}$ (state space and action space) are compact subsets of $\mathbb{R}^{\mathcal{D}_s}$ and $\mathbb{R}^{\mathcal{D}_a}$ respectively.

2. All the rewards are bounded such that $r(s, a) \in [R_{\min}, R_{\max}]$, for all $(s, a) \in \mathcal{S} \times \mathcal{A}$.

3. The optimal $Q^*$ is $C$-Lipschitz such that

$$\left|Q^*(s, a) - Q^*(s', a')\right| \leq C\left(||s - s'||_F + ||a - a'||_F\right)$$

Now we prove some useful lemmas for proving sampling complexity of Hamiltonian $Q$-Learning

**Lemma 1.** *For some constant $c_1$, if*

$$|\Omega_t| \geq c_1 \frac{\max\left\{|\mathcal{S}|^2, |\mathcal{A}|^2\right\}|\mathcal{S}||\mathcal{A}|\mathcal{D}_s\mathcal{D}_a}{\log(\mathcal{D}_s + \mathcal{D}_a)}$$

*with $\left\|\widehat{Q}^t(s, a) - Q^*(s, a)\right\|_\infty \leq \epsilon$ then there exists a constant $c_2$ such that*

$$\left\|Q^t(s, a) - Q^*(s, a)\right\|_\infty \leq c_2\epsilon$$

**Proof of Lemma 1.** Recall that in order to complete a low rank matrix using matrix estimation methods, the matrix can not be sparse. This condition can be formulated using the notion of incoherence. Let $Q$ be a matrix of rank $r_Q$ with the singular value decomposition $Q = U\Sigma V^T$. Let $T_Q$ be the orthogonal projection of $Q \in \mathbb{R}^{|\mathcal{S}|\times|\mathcal{A}|}$ to its column space. Then incoherence parameter of $\phi(Q)$ can be give as

$$\phi(Q) = \max\left\{\frac{|\mathcal{S}|}{r_Q} \max_{1\leq i\leq|\mathcal{S}|} ||T_U\mathbf{e}_i||_F^2, \frac{|\mathcal{A}|}{r_Q} \max_{1\leq i\leq|\mathcal{A}|} ||T_U\mathbf{e}_i||_F^2\right\}$$

where $\mathbf{e}_i$ are the standard basis vectors. Recall that $Q^t$ is the matrix generated in matrix completion phase from $\widehat{Q}$. From Theorem 4 in Chen & Chi (2018) we have that for some constant $C_1$ if a fraction of $p$ elements are observed from the matrix such that

$$p \geq C_1\frac{\phi_t^2 r_Q^2 \mathcal{D}_s\mathcal{D}_a}{\log(\mathcal{D}_s + \mathcal{D}_a)}$$

where $\phi_t$ is the coherence parameter of $Q^t$ then with probability at least $1 - C_2(\mathcal{D}_s + \mathcal{D}_a)^{-1}$ for some constant $C_2$ with $\left\|\widehat{Q}^t(s, a) - Q^*(s, a)\right\|_\infty \leq \epsilon$ there exists a constant $c_2$ such that

$$\left\|Q^t(s, a) - Q^*(s, a)\right\|_\infty \leq c_2\epsilon$$

Note that $p \approx \frac{|\Omega_t|}{|\mathcal{S}||\mathcal{A}|}$. Further we have for some constant $c_3$

$$\frac{\phi_t^2 r_Q^2 \mathcal{D}_s\mathcal{D}_a}{\log(\mathcal{D}_s + \mathcal{D}_a)} = c_3 \frac{\max\left\{|\mathcal{S}|^2, |\mathcal{A}|^2\right\}\mathcal{D}_s\mathcal{D}_a}{\log(\mathcal{D}_s + \mathcal{D}_a)}$$

Thus it follows that for some constant $c_1$ if

$$|\Omega_t| = c_1 \frac{\max\left\{|\mathcal{S}|^2, |\mathcal{A}|^2\right\}|\mathcal{S}||\mathcal{A}|\mathcal{D}_s\mathcal{D}_a}{\log(\mathcal{D}_s + \mathcal{D}_a)}$$

with $\left\|\left\|\widehat{Q}^t(s,a) - Q^*(s,a)\right\|\right\|_\infty \le \epsilon$ then there exists a constant $c_2$ such that

$$\left\|\left\|Q^t(s,a) - Q^*(s,a)\right\|\right\|_\infty \le c_2\epsilon$$

This concludes the proof of Lemma 1. □

**Lemma 2.** *Let $1 - \xi$ be the spectral gap of Markov chain under Hamiltonian sampling where $\xi \in [0,1]$. Let $\Delta R = R_{\max} - R_{\min}$ be the maximum reward gap. Then $\forall (s',a') \in \mathcal{S} \times \mathcal{A}$ we have that*

$$|\widehat{Q}(s',a') - Q^*(s',a')| \le \frac{\gamma^2}{1-\gamma}\Delta R + \sqrt{\frac{1+\xi}{1-\xi}\frac{2}{|\mathcal{H}|}\left(\frac{\gamma R_{\max}}{1-\gamma}\right)^2\log\left(\frac{2}{\delta}\right)}.$$

*with at least probability $1 - \delta$.*

**Proof of Lemma 2.** Let $\widehat{Q}(s',a') = r(s',a') + \frac{\gamma}{|\mathcal{H}|}\sum_{s\in\mathcal{H}}\max_a Q(s,a)$. Recall that $\mathcal{M}Q(s) = \max_a Q(s,a)$. Then we have that $\widehat{Q}(s',a') = r(s',a') + \frac{\gamma}{|\mathcal{H}|}\sum_{s\in\mathcal{H}}\mathcal{M}Q(s)$. Then it follows that

$$|\widehat{Q}(s',a') - Q^*(s',a')| = \left|r(s',a') + \frac{\gamma}{|\mathcal{H}|}\sum_{s\in\mathcal{H}}\mathcal{M}Q(s) - r(s',a') - \gamma\mathbb{E}_{\mathcal{P}}\mathcal{M}Q^*(s)\right|$$

$$= \left|\frac{\gamma}{|\mathcal{H}|}\sum_{i=1}^{|\mathcal{H}|}\mathcal{M}Q(s_i) - \gamma\mathbb{E}_{\mathcal{P}}\mathcal{M}Q^*(s)\right|$$

$$= \left|\frac{\gamma}{|\mathcal{H}|}\sum_{i=1}^{|\mathcal{H}|}\mathcal{M}Q(s_i) - \frac{\gamma}{|\mathcal{H}|}\sum_{i=1}^{|\mathcal{H}|}\mathcal{M}Q^*(s_i)\right|$$

$$+ \left|\frac{\gamma}{|\mathcal{H}|}\sum_{i=1}^{|\mathcal{H}|}\mathcal{M}Q^*(s_i) - \gamma\mathbb{E}_{\mathcal{P}}\mathcal{M}Q^*(s)\right| \tag{16}$$

Recall that all the rewards are bounded such that $r(s,a) \in [R_{\min}, R_{\max}]$, for all $(s,a) \in \mathcal{S} \times \mathcal{A}$. Thus for all $s, a$ we have that $\mathcal{M}Q(s) \le \frac{\gamma}{1-\gamma}R_{\max}$. Let $\Delta R = R_{\max} - R_{\min}$. Then we have that

$$\left|\frac{\gamma}{|\mathcal{H}|}\sum_{i=1}^{|\mathcal{H}|}\mathcal{M}Q(s_i) - \frac{\gamma}{|\mathcal{H}|}\sum_{i=1}^{|\mathcal{H}|}\mathcal{M}Q^*(s_i)\right| \le \frac{\gamma^2}{1-\gamma}\Delta R. \tag{17}$$

Let $\xi \in [0,1]$ be a constant such that $1 - \xi$ is the spectral gap of the Markov chain under Hamiltonian sampling. Then from Fan et al. (2018) we have that

$$\mathbb{P}\left(\frac{1}{|\mathcal{H}|}\sum_{i=1}^{|\mathcal{H}|}\mathcal{M}Q^*(s_i) - \mathbb{E}_{\mathcal{P}}\mathcal{M}Q^*(s) \ge \vartheta\right) \le \exp\left(-\frac{1-\xi}{1+\xi}\frac{|\mathcal{H}|\vartheta^2}{2R_{\max}^2}\left(\frac{1-\gamma}{\gamma}\right)^2\right)$$

Let $\delta = \exp\left(-\frac{1-\xi}{1+\xi}\frac{|\mathcal{H}|\vartheta^2}{2R_{\max}^2}\left(\frac{1-\gamma}{\gamma}\right)^2\right)$. Then we have that

$$\vartheta = \sqrt{\frac{1+\xi}{1-\xi}\frac{2}{|\mathcal{H}|}\left(\frac{\gamma R_{\max}}{1-\gamma}\right)^2\log\left(\frac{2}{\delta}\right)}.$$

Thus we see that

$$\left|\frac{1}{|\mathcal{H}|}\sum_{i=1}^{|\mathcal{H}|}\mathcal{M}Q^*(s_i) - \mathbb{E}_{\mathcal{P}}\mathcal{M}Q^*(s)\right| \le \sqrt{\frac{1+\xi}{1-\xi}\frac{2}{|\mathcal{H}|}\left(\frac{\gamma R_{\max}}{1-\gamma}\right)^2\log\left(\frac{2}{\delta}\right)} \tag{18}$$

with at least probability $1 - \delta$. Thus it follows from equations equation 16, equation 17 and equation 18 that

$$|\widehat{Q}(s',a') - Q^*(s',a')| \le \frac{\gamma^2}{1-\gamma}\Delta R + \sqrt{\frac{1+\xi}{1-\xi}\frac{2}{|\mathcal{H}|}\left(\frac{\gamma R_{\max}}{1-\gamma}\right)^2\log\left(\frac{2}{\delta}\right)}.$$

with at least probability $1 - \delta$. This concludes the proof of Lemma 2. □

**Lemma 3.** *For all $(s,a) \in \mathcal{S} \times \mathcal{A}$ we have that*

$$|Q^t(s,a) - Q^*(s,a)| \le 2c_1\frac{\gamma^2 R_{\max}}{1-\gamma}$$

*with probability at least $1 - \delta$*

**Proof of Lemma 3.** From Lemma 2 and Shah et al. (2020) we have that for all $(s, a) \in \Omega_t$

$$\left|\widehat{Q}^t(s, a) - Q^*(s, a)\right| \leq \frac{\gamma^2}{1 - \gamma} \Delta R + \sqrt{\frac{1 + \xi}{1 - \xi} \frac{2}{|\mathcal{H}_t|} \left(\frac{\gamma R_{\max}}{1 - \gamma}\right)^2 \log\left(\frac{2|\Omega_t|T}{\delta}\right)}. \quad (19)$$

with probability at least $1 - \frac{\delta}{T}$. Thus we have that

$$\left|Q^t(s, a) - Q^*(s, a)\right| \leq c_1 \frac{\gamma^2}{1 - \gamma} \Delta R + c_1 \sqrt{\frac{1 + \xi}{1 - \xi} \frac{2}{|\mathcal{H}_t|} \left(\frac{\gamma R_{\max}}{1 - \gamma}\right)^2 \log\left(\frac{2|\Omega_t|T}{\delta}\right)}.$$

with probability at least $1 - \frac{\delta}{T}$. Fro all $1 \leq t \leq T$ letting

$$|\mathcal{H}_t| = \frac{1 + \xi}{1 - \xi} \frac{2}{\gamma^2} \log\left(\frac{2|\Omega_t|T}{\delta}\right)$$

we obtain

$$\frac{\gamma^2}{1 - \gamma} R_{\max} \geq \sqrt{\frac{1 + \xi}{1 - \xi} \frac{2}{|\mathcal{H}_t|} \left(\frac{\gamma R_{\max}}{1 - \gamma}\right)^2 \log\left(\frac{2|\Omega_t|T}{\delta}\right)}.$$

Thus we have,

$$\left|Q^t(s, a) - Q^*(s, a)\right| \leq 2c_1 \frac{\gamma^2 R_{\max}}{1 - \gamma}$$

with probability at least $1 - \delta$. Recall that $\forall (s, a) \in \mathcal{S} \times \mathcal{A}$ we have $\mathcal{M}Q(s, a) \leq \frac{\gamma \Delta R}{1 - \gamma}$. Thus this also proves that

$$\left|Q^t(s, a) - Q^*(s, a)\right| \leq 2c_1 \gamma |Q^{t-1}(s, a) - Q^*(s, a)|$$

This concludes the proof of Lemma 3. $\square$

Now we proceed to prove the main theorem for sampling complexity as follows.

**Theorem 3.** *Let $\mathcal{D}_s, \mathcal{D}_a$ be the dimension of state space and action space respectively. Consider the Hamiltonian Q-Learning algorithm presented in Algorithm 1. Under a suitable matrix completion method sampling complexity of the algorithm, Q function converge to the family of $\epsilon$-optimal Q functions with $\widetilde{O}\left(\epsilon^{-(\mathcal{D}_s + \mathcal{D}_a + 2)}\right)$ number of samples.*

**Proof of Theorem 3.** Note that sample complexity of Hamiltonian $Q$-Learning can be given as

$$\sum_{t=1}^{T_\epsilon} |\Omega_t||\mathcal{H}_t| \leq T_\epsilon |\Omega_{T_\epsilon}||\mathcal{H}_{T_\epsilon}|$$

Let $\beta^t$ be the discretization parameter at time $t$ and $T_\epsilon = \frac{\log\left(\frac{\gamma R_{\max}}{(1 - \gamma)\epsilon}\right)}{\log\left(\frac{1}{2\gamma c_1}\right)}$. Then from Lemmas 1, 2 and 3 it follows that

$$\sum_{t=1}^{T_\epsilon} |\Omega_t||\mathcal{H}_t| = \widetilde{O}\left(\frac{1}{\epsilon^{\mathcal{D}_s + \mathcal{D}_a + 2}}\right)$$

This concludes the proof of Theorem 3. $\square$

## C ADDITIONAL EXPERIMENTAL DETAILS AND RESULTS FOR CART-POLE

Let $\theta, \dot{\theta}$ be the angle and angular velocity of the pole, respectively. Let $x, \dot{x}$ be the position and linear velocity of the cart, respectively. Let $a$ be the control force applied to the cart. Then, by defining $m$, $M$, $l$ and $g$ as the mass of the pole, mass of the cart, length of the pole and gravitational acceleration, respectively, the dynamics of cart-pole system can be expressed as

$$\ddot{\theta} = \frac{g \sin\theta - \frac{a + ml\dot{\theta}^2 \sin\theta}{m + M} \cos\theta}{l\left(\frac{4}{3} - \frac{m \cos^2\theta}{m + M}\right)}$$

$$\ddot{x} = \frac{a + ml\left(\dot{\theta}^2 \sin\theta - \ddot{\theta} \cos\theta\right)}{m + M}$$

(20)

State space of cart-pole system is 4-dimensional ($\mathcal{D}_s = 4$) and any state $s \in \mathcal{S}$ is given by $s = (\theta, \dot{\theta}, x, \dot{x})$. We define the range of state space as $\theta \in [-pi/2, \pi/2], \dot{\theta} \in [-3.0, 3.0], x \in [-2.4, 2.4]$ and $\dot{x} \in [-3.5, 3.5]$. We consider action space to be a 1-dimensional ($\mathcal{D}_a = 1$) space such that $a \in [-10, 10]$. We discretize each dimension in state space into 5 values and action space into 10 values. This forms a $Q$ matrix of dimensions $625 \times 10$.

Although the differential equations (20) governing the dynamics of the pendulum on a cart system are deterministic, uncertainty of the parameters and external disturbances to the system causes the cart pole to deviate from the defined dynamics leading to a stochastic state transition. Following the conventional approach we model these parameter uncertainties and external disturbances using a multivariate Gaussian perturbation (Maithripala et al., 2016; Madhushani et al., 2017; McAllister & Rasmussen, 2017). Here we consider the co-variance of the Gaussian perturbation to be $\Sigma = \text{diag}[0.143, 0.990, 0.635, 1.346]$.

Let $s_t = (\theta_t, \dot{\theta}_t, x_t, \dot{x}_t)$ and $a_t$ be the state and the action at time $t$. Then the state transition probability kernel and corresponding target distribution can be given using (7) and (8), respectively, with mean $\mu(s_t, a_t) = (\theta_t + \dot{\theta}_t\tau, \dot{\theta}_t + \ddot{\theta}_t\tau, x_t + \dot{x}_t\tau, \dot{x}_t + \ddot{x}_t\tau)$, where $\ddot{\theta}_t, \ddot{x}_t$ can be obtained from (20) by substituting $\theta_t, \dot{\theta}_t, a_t$, and co-variance $\Sigma(s_t, a_t) = \Sigma$.

Our simulation results use the following value for the system parameters - $m = 0.1kg$, $M = 1kg$, $l = 0.5m$ and $g = 9.8ms^{-2}$. We take 100 HMC samples during the update phase. We use trajectory length $L = 100$ and step size $\delta l = 0.02$. We randomly initialize the $Q$ matrix using values between 0 and 1. We provide additional comparison heat maps for first two dimensions $\theta, \dot{\theta}$ with fixed $x, \dot{x}$.

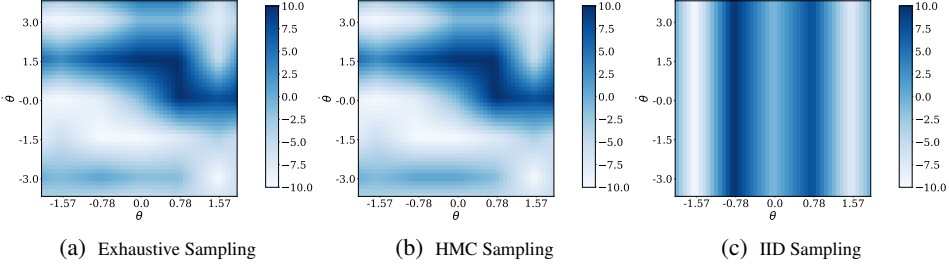

(a) Exhaustive Sampling    (b) HMC Sampling    (c) IID Sampling

Figure 6: Figure 6(a), 6(b) and 6(c) show policy heat maps for $Q$-Learning with exhaustive sampling, Hamiltonian $Q$-Learning and $Q$-Learning with IID sampling, respectively.

Further, we provide additional comparison heat maps for last two dimensions $x, \dot{x}$ with fixed $\theta, \dot{\theta}$.

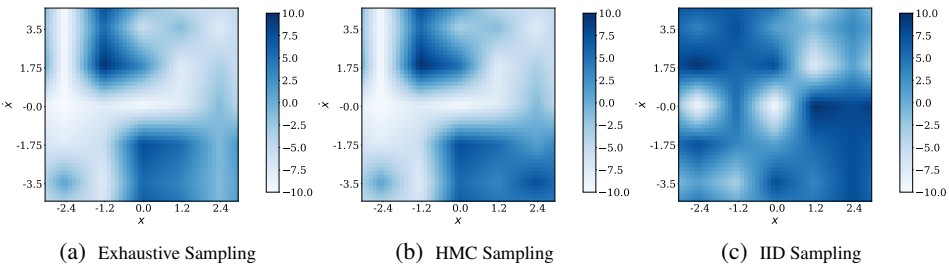

(a) Exhaustive Sampling    (b) HMC Sampling    (c) IID Sampling

Figure 7: Figure 7(a), 7(b) and 7(c) show policy heat maps for $Q$-Learning with exhaustive sampling, Hamiltonian $Q$-Learning and $Q$-Learning with IID sampling, respectively.

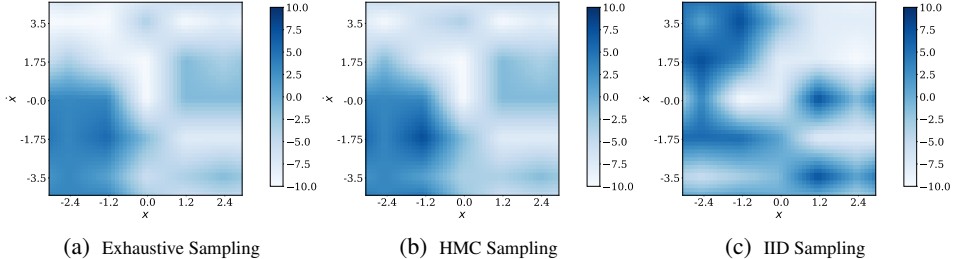

(a) Exhaustive Sampling    (b) HMC Sampling    (c) IID Sampling

Figure 8: Figure 8(a), 8(b) and 8(c) show policy heat maps for $Q$-Learning with exhaustive sampling, Hamiltonian $Q$-Learning and $Q$-Learning with IID sampling, respectively.

## D  ADDITIONAL DETAILS FOR OCEAN SAMPLING APPLICATION

**Glider dynamics**  By assuming that the glider's motion is restricted to an horizontal plane (Refael & Degani, 2019), we let $x$, $y$ and $\theta$ denote its center of mass position and heading angle, respectively. Then we can define the 6-dimensional state vector for this system as $s = (x, y, \dot{x}, \dot{y}, \theta, \dot{\theta})$ and the action $a$ as a scalar control input to the glider. Also, to accommodate dynamic perturbations due to the ocean current, other external disturbances and parameter uncertainties, we assume that the probabilistic state transition is governed by a multivariate Gaussian. We consider that the motion of the glider is restricted to a horizontal plane. Let $x$, $y$ and $\theta$ be the coordinates of the center of mass of glider and heading angle respectively. By introducing $q = \begin{bmatrix} x & y & \theta \end{bmatrix}^T$, the dynamics of the glider can be expressed as

$$M\ddot{q} = RF_f + F_b + \tau$$

where

$$M = diag \begin{bmatrix} m \\ m \\ \mathcal{I}_{in} + \mathcal{I}_{out} \end{bmatrix}; \quad R = \begin{bmatrix} \cos\theta & -\sin\theta & 0 \\ \sin\theta & \cos\theta & 0 \\ 0 & 0 & 1 \end{bmatrix}$$

$$F_f = \begin{bmatrix} \alpha_f \dot{\theta}^2 sgn(\dot{\theta}) \sin(\beta + \psi) \\ \alpha_f \dot{\theta}^2 \cos(\beta + \psi) \\ 0 \end{bmatrix}; \quad F_b = -\alpha_b \sqrt{\dot{x}^2 + \dot{y}^2} \begin{bmatrix} \dot{x} \\ \dot{y} \\ 0 \end{bmatrix}; \quad \tau = \begin{bmatrix} 0 \\ 0 \\ -\mu_f sgn(\dot{\theta})\dot{\theta}^2 - \mathcal{I}_{in}a \end{bmatrix}$$

$$\alpha_f = \frac{1}{2}\rho C_f d_f L \left( r^2 + \left(\frac{L}{2}\right)^2 + rL \cos\psi \right); \quad \mu_f = \alpha_f \left( \frac{L}{2} + r \cos\psi \right); \quad \alpha_b = \frac{1}{2}C_b d_b \pi r$$

$$\alpha_b = 0.005; \quad \alpha_f = 0.062; \quad \mu_f = 0.0074; \quad \sigma = 2.5.$$

Our simulation results use system parameter values from Table 1. We define the range of state and action space as $x, y \in [-10, 10]$, $\dot{x}, \dot{y} \in [-25, 25]$, $\theta \in [-\pi, \pi]$, $\dot{\theta} \in [-3, 3]$, and $a \in [-1, 1]$, respectively and then discretizing each state dimension into 5 distinct values and the action space into 5 distinct values, we have a $Q$ matrix of size $15625 \times 5$. Also, we assume that the state transition kernel is given by a multivariate Gaussian with zero mean and covariance $\Sigma = \text{diag}[11.111, 69.444, 11.111, 69.444, 0.143, 0.990]$. After initializing the $Q$ matrix using randomly chosen values from $[0, 1]$, we sample state-action pairs independently with probability $p = 0.5$ at each iteration. Also, we assume $\sigma = 2.5$, $\lambda = 0.1$, $\mathcal{C} = \text{diag}[1, 0]$. We take 100 HMC samples during the update phase. We use trajectory length $L = 100$ and step size $\delta l = 0.02$.

**Additional experimental results**  :  We provide additional comparison heat maps for first two dimensions $x, y$ with fixed $\dot{x}, \dot{y}, \theta, \dot{\theta}$.

Table 1: Notations

| Notations | Description | Value |
|---|---|---|
| $m$ | total mass | $1.03 \ kg$ |
| $\mathcal{I}_{in}$ | inner body moment of inertia | $0.5 \ kg \ m^2$ |
| $\mathcal{I}_{out}$ | outer body moment of inertia | $0.174 \ kg \ m^2$ |
| $r$ | radius | $0.08 \ m$ |
| $L$ | length of the flap | $0.09 \ m$ |
| $d_f$ | submersible depth of the flap | $0.044 \ m$ |
| $d_b$ | submersible depth of the body | $0.02 \ m$ |
| $\beta$ | angular location of the flap | $30^o$ |
| $\psi$ | maximum open angle of the flap | $20^o$ |
| $C_f$ | drag coefficient of the flap | $2$ |
| $C_b$ | drag coefficient of the body | $2$ |
| $\rho$ | water density | $1027 \ kg \ m^3$ |

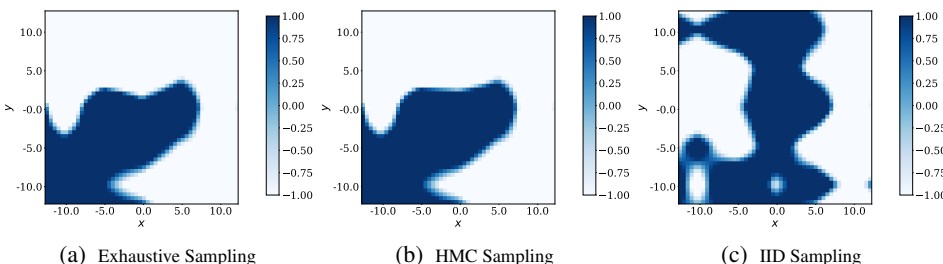

(a) Exhaustive Sampling     (b) HMC Sampling     (c) IID Sampling

Figure 9: Figure 9(a), 9(b) and 9(c) show policy heat maps for $Q$-Learning with exhaustive sampling, Hamiltonian $Q$-Learning and $Q$-Learning with IID sampling, respectively.

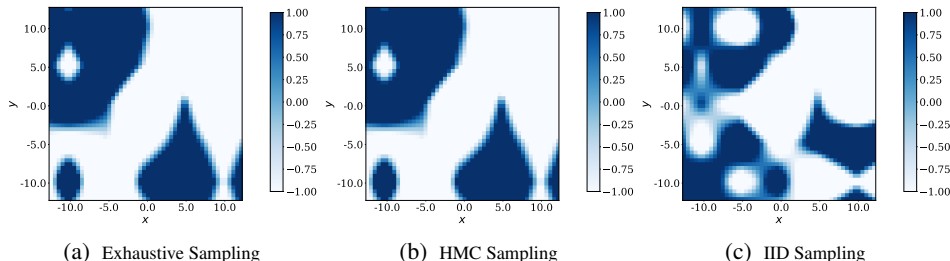

(a) Exhaustive Sampling     (b) HMC Sampling     (c) IID Sampling

Figure 10: Figure 10(a), 10(b) and 10(c) show policy heat maps for $Q$-Learning with exhaustive sampling, Hamiltonian $Q$-Learning and $Q$-Learning with IID sampling, respectively.

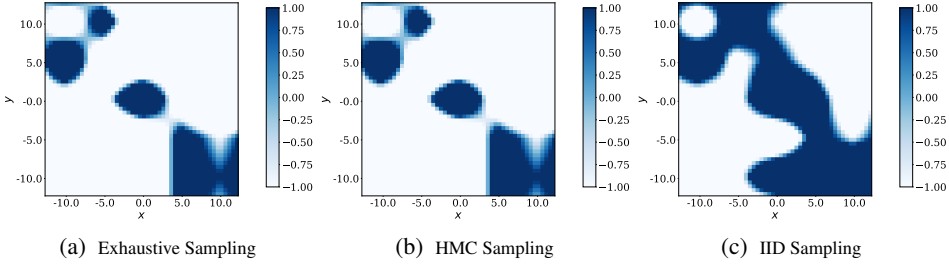

(a) Exhaustive Sampling     (b) HMC Sampling     (c) IID Sampling

Figure 11: Figure 11(a), 11(b) and 11(c) show policy heat maps for $Q$-Learning with exhaustive sampling, Hamiltonian $Q$-Learning and $Q$-Learning with IID sampling, respectively.

# E   APPLICATION OF HMC SAMPLING FOR $Q$-LEARNING: FURTHER DETAILS

In this section we provide a detailed explanation on drawing HMC samples from a given state transition probability distribution. Let $(s_t, a_t)$ be the current state action pair. Let $\mu(s_t, a_t), \Sigma(s_t, a_t)$ be the mean and covariance of the transition probability kernel. In order to draw HMC samples

we are required to define the corresponding potential energy and kinetic energy of the system. Let $\mathcal{P}(s|s_t, a_t)$ be the smooth target state transition distribution.

**Potential energy, kinetic energy and mass:**   In this work we consider $\mathcal{P}(s|s_t, a_t)$ to be a truncated multivariate Gaussian as given in equation 8. Thus potential energy can be explicitly given as,

$$U(s) = -\log(\mathcal{P}(s)) = \frac{1}{2}(s-\mu)^T \Sigma^{-1}(s-\mu) - \frac{1}{2}\log\left((2\pi)^{D_s}\det(\Sigma)\right)$$

$$-\sum_{i=1}^{D_s}\left[\log\left(1 + \exp(-\kappa(d_i^+ - s^i))\right) + \log\left(1 + \exp(-\kappa(s^i - d_i^-))\right)\right]$$

where, $\mu$ and $\Sigma$ correspond to the mean and variance of the transition probability kernel. In the context of HMC $s$ is referred to as the position variable. Then we chose kinetic energy can be given as

$$K(v) = -\log(\mathcal{P}(v|s)) = \frac{1}{2}v^T M^{-1} v = \frac{1}{2}v^T \Sigma v.$$

where $v$ is the momentum variable and $M = \Sigma^{-1}$ corresponds to the mass/inertia matrix associated with the Hamiltonian.

**Hamiltonian Dynamics:**   As the Hamiltonian is the sum of the kinetic and the potential energy, i.e. $H(s,v) = U(s) + K(v)$, the Hamiltonian dynamics can be expressed as

$$\dot{s} = \frac{\partial K}{\partial v} = \Sigma v$$

and

$$\dot{v} = -\frac{\partial U}{\partial s} = -\Sigma^{-1}(s-\mu) + \kappa\left[S(-\kappa(d^+ - s)) - S(-\kappa(s - d^-))\right],$$

where $S(\xi_1, \cdots, \xi_{D_s}) = \left[S(\xi_1), \cdots, S(\xi_{D_s})\right]^T$ denotes element wise sigmoid function of the vector $\xi$. We initialize HMC sampling by drawing a random sample $s$ from the transition probability distribution and a new momentum variable $v$ from the multivariate Gaussian $\mathcal{N}(0, \Sigma^{-1})$. We integrate the Hamiltonian dynamics for $L$ steps with step size $\Delta l$ to generate the trajectory from $(s, v)$ to $(s', v')$. To ensure that the Hamiltonian is conserved along the trajectory, we use a volume preserving symplectic integrator, in particular a leapfrog integrator which uses the following update rule to go from step $l$ to $l + \Delta l$:

$$v_{l+\frac{\Delta l}{2}} = v_l - 0.5\Delta l \left.\frac{\partial U(s)}{\partial s}\right|_l, \quad s_{l+\Delta l} = s_l + \Delta l \Sigma v_{l+\frac{\Delta l}{2}}, \quad v_{l+\Delta l} = v_l - 0.5\Delta l \left.\frac{\partial U(s)}{\partial s}\right|_{l+\Delta l}.$$

**Acceptance of the new proposal:**   Then, following the Metropolis–Hastings acceptance/rejection rule, this new proposal is accepted with probability

$$\min\left\{1, \frac{\exp\left(H(s, v)\right)}{\exp\left(H(s', -v')\right)}\right\}.$$

**Updating $Q$ function using HMC samples:**   Let $\mathcal{H}_t$ be the set of next states obtained via HMC sampling, i.e., position variables from the accepted set of proposals. Then we update $Q(s_t, a_t)$ using equation 9.

