# OpenReview forum: "Hamiltonian Q-Learning: Leveraging Importance-sampling for Data Efficient RL"
_ICLR.cc/2021/Conference — Reject_

### Official Review · AnonReviewer4 · 2020-10-13
**This paper studies appling Hamiltonian sampling to computing Bellman equation approximately. I have a few questions:**

**Rating:** 6
**Confidence:** 3

**Review:**

This paper studies appling Hamiltonian sampling to computing Bellman equation (thus Q-learning iterations) approximately. I have a few questions:

 1. The Hamiltonian sampling approach requires knowing the distribution in advance. How is this possible in our setting? If we are estimating this distribution, then since it is high dimensional, we still need large amount of data. This is the most confusing point from my point of view.
2. As for the matrix completion approach: why is this reasonable? Why does the Q-table have low-rank property? I think at least this assumption should be justified empirically, otherwise, the theoretical result doesn't make any sense.  Also, the definition of the low-rank property itself is very suspicious, because it needs to hold for any policy. As a result, this may be very hard to verify rigorously. But at least some preliminary evidence should be presented.

---

> ### Author Response · Authors · 2020-11-22
> **Thanks so much for your feedback! In this revised version, we have emphasized the justification for assuming low-rank structure.**
>
> Thank you for your feedback and insightful comments! In the following, we address your concerns individually.
>
> --- **Prior Knowledge about the distribution:** We only assume that the state transition distribution is known up to a constant; we do not estimate it. For classical control tasks, stochasticity in the state transition arises from parameter uncertainties, measurement noise and external disturbances - thus it is reasonable to assume that we know the state transition probability distribution up to a constant.
>
> --- **Assumption on Low-rank Structure of the $Q$ matrix:** This assumption exploits the fact that the underlying state transition and reward function are endowed with some structure in a large class of problems. Prior work on value function approximation based approaches for reinforcement learning also leveraged a similar assumption implicitly while using various basis functions (e.g. CMAC, radial basis functions, diffusion wavelets etc.).  More recently, Yang et. al. ("Harnessing Structures for Value-Based Planning and Reinforcement Learning", ICLR 2020) have empirically shown that the $Q$-matrices for benchmark Atari games and classical control tasks exhibit low-rank structure. We have included a brief description of the rationale behind this assumption in Section 2.3 (Low-rank Structure in $Q$-learning and Matrix Completion) of our revised manuscript. Furthermore, the definition of the low rank structure does not depend on the policy; we can use exhaustive sampling to calculate optimal $Q^*$ and examine the low rank structure of that matrix. In particular, the rank depends only on the state transition matrix and the reward function, not on the policy itself.

---

> > ### Comment · AnonReviewer4 · 2020-11-23
> > **Prior Knowledge about the distribution**
> >
> > Thanks for your explanation. Since we assume the state transition is known, in principle the MDP itself is known, right? If that is the case, from my point of view, the advantage of the sampling approach is to reduce computation instead of data required?

---

> > > ### Author Response · Authors · 2020-11-23
> > > **Thank you for your quick response! We only assume partial knowledge about the underlying MDP.**
> > >
> > > Since we assume that the state transition probability distribution is known only up to a constant and the deterministic dynamics of the system is unknown, MDP is only partially known. We reduce the computations by reducing the number of samples required to estimate the $Q$ function. We have provided theoretical guarantees for sample efficiency in Theorem 3. As illustrated in Figures 3 and 4, Hamiltonian $Q$ learning converges to an $\epsilon$ optimal $Q$ function with a significantly smaller number of samples compared to exhaustive sampling. As a result, Hamiltonian $Q$ learning requires a significantly small amount of data to achieve the same accuracy as exhaustive sampling. Thus we reduce the computations by reducing the number of samples used in $Q$ value estimation.

---

> > > > ### Comment · AnonReviewer4 · 2020-11-23
> > > > **Thanks for your explanation. I have updated my score.**
> > > >
> > > > Thanks for your explanation. I have updated my score.

---

> > > > > ### Author Response · Authors · 2020-11-25
> > > > > **Thank you for your response**
> > > > >
> > > > > Thank you for increasing the score.

---

### Official Review · AnonReviewer3 · 2020-10-27
**Convergence contributions (Theorems 1-2) do not exhibit gains over vanilla Q learning.**

**Rating:** 5
**Confidence:** 4

**Review:**


Strengths:

1) To my knowledge, this is the first credible effort to apply Hamiltonian Monte Carlo to Q learning in order to avoid some degree of the random sampling required for its almost sure convergence. At least the algorithm novelty is apparent.

2) The experiments clearly demonstrate the merits of the proposed importance sampling scheme for obtaining improved convergence of Q learning, and for yielding coverage of the space comparable to exhaustive sampling.

3) The idea of using importance sampling *during* training is a key distinguishing feature from a majority of other works which use it for, upon the basis of a fixed prior policy, improving the current policy.



Weaknesses:


1) The preliminaries section is disjoint/fragmented. The limitations of Q learning (equation (1)) should directly motivate use of Hamiltonian Monte Carlo (HMC) in Section 2.2, but instead the manner in which HMC is presented is as additional preliminary material. This is not inherently disqualifying, but a number of exotic terms are introduced in the ``preliminaries" Section 2.2 without explanation or otherwise connection to the RL problem, such as momentum variable, leapfrog integrator, Hamiltonian, etc. Why are these entities pertinent to MDPs and Q learning? This needs to be more carefully and conscientiously connected. A similar comment is true for Section 2.3 -- matrix completion or the need for addressing matrix estimation problems is nowhere to be found in 2.1 and 2.2, which leaves the reader confused as to why matrix completion is being discussed. This overall disjointedness then makes the conceptual innovation of this work more mysterious to understand, which is a concern.

2) The reasoning in the paragraph before Section 3.1 seems incorrect. The fact that samples in a limited pool of IID samples concentrate around the region with high
probability density is not evidence that this is a poor estimate for the expected value, but rather that the expected value is not representative of a uniform distribution across  the space. Thus, I think the authors might have intended to  be talking about higher-order moments of this conditional distribution, such as the skewness, kurtosis, etc. or otherwise risk measures such as CVaR.

3) The main convergence theory (Theorems 1-2) do not exhibit any discernible complexity or sample efficiency gains over vanilla Q learning. Moreover, a coherent discussion of the technical innovations required to establish these theorems is absent from the manuscript. These limitations alone are very concerning.

4) The matrix completion step (equations (9)-(10) is playing the role of a proximal operator on the Q learning update, or otherwise some projection of the Bellman error onto a low-dimensional subspace of features. Therefore, the steps conducted in Section 3.2 are very similar mathematically to entropic regularization, which may also be seen as a special case of a proximal operator on the space of Q functions. This connection is not made in the paper, as well as the more computational/statistical motivation for where the matrix competition step comes from. I strongly suggest the authors consider better explaining the links between step (4) of Algorithm 1 and proximal methods in any revision of this work.

5) Proofs seem fairly routine in my reading. What is new or innovative here? Again, this is not explained anywhere.


Minor Comments:


1) References missing on distributional/representational aspects of Q learning:

Dearden, R., Friedman, N., & Russell, S. (1998, July). Bayesian Q-learning. In AAAI (pp. 761-768).

Koppel, A., Tolstaya, E., Stump, E., & Ribeiro, A. (2018). Nonparametric stochastic compositional gradient descent for q-learning in continuous markov decision problems. arXiv preprint arXiv:1804.07323.

Jeong, H., Zhang, C., Pappas, G. J., & Lee, D. D. (2019, August). Assumed density filtering Q-learning. In Proceedings of the 28th International Joint Conference on Artificial Intelligence (pp. 2607-2613). AAAI Press.

---

> ### Author Response · Authors · 2020-11-22
> **Thanks so much for your feedback! We have added theoretical and experimental results on sampling complexity.**
>
> Thank you for your feedback and insightful comments! In the following, we address your concerns individually.
>
> --- **Preliminaries Section:** We have made significant changes in Section 2 to improve its readability as well as to connect it with the subsequent sections. After providing a brief discussion on MDP and $Q$ learning, we first identify the need for an efficient algorithm before introducing the concept of HMC sampling in this revised version. We have also reorganized Section 2.2 to keep the discussion on HMC more focussed and connected with the rest of the paper. In Section 2.3, we have also added a short discussion to highlight the motivation behind assuming the low-rank structure; we expect that this discussion would be helpful in emphasizing the relevance of matrix completion techniques in this context.
>
> --- **Reasoning related to IID Sampling:** We assume that the distribution is known only upto a constant. Drawing IID samples from such distributions, especially when they are high-dimensional, is computationally very expensive. This motivates the use of HMC sampling for estimating the expected $Q$-value associated with the next states. In this revised version, we have edited and rephrased the reasoning in the paragraph before Section 3.1 to make this aspect less ambiguous.
>
> --- **Theoretical Results and their Novelty:** In addition to our initial theoretical results on convergence guarantees under HMC sampling and boundedness guarantees under matrix completion with HMC sampling, we have incorporated an theoretical result (Theorem 3) on sampling complexity of our algorithm in the revised version. We show that sampling complexity of our algorithm matches the lower bound of min-max sampling complexity. Also, to improve the flow of Section 3.3 and clarity of the contribution of this work, we have added brief descriptions to complement the mathematical details in Theorem 1, 2, and 3. Please see Section 3.3 and Appendix B for the related changes. To the best of our knowledge this is the first work to show convergence of $Q$ function under Hamiltonian sampling. This is the first work to analyze the sampling complexity under HMC sampling with matrix completion.
>
> --- **Connection with Proximal Methods:** We apologize for not addressing the connection of our method to proximal methods in this revision. We will add this discussion in the 2nd revision that we will upload within the next two days.
>
> --- **Minor Comments:** We have added the missing references in the related work section of the revised paper.

---

> ### Author Response · Authors · 2020-11-24
> **We have discussed the connection to entropy regularization**
>
> We have uploaded a revised version of the paper which cites relevant references on using entropy regularization for Reinforcement training. However, our approach is different from entropy regularization methods which promote sparsity in the learned $Q$ function. In this work, we instead focus on promoting low-rank $Q$ matrices and formulate a corresponding constrained optimization problem to recover the $Q$ matrix from a small subset of its elements which have been updated using the Bellman equation.

---

### Official Review · AnonReviewer2 · 2020-10-28
**Interesting title but contributions are not convincing**

**Rating:** 6
**Confidence:** 4

**Review:**

In this paper, a Hamiltonian Q-learning is proposed by combining Hamiltonian Monte Carlo with matrix completion. Evaluations are compared with Q-learning.

1. How to interpret Eqn.(3)(9) as a Hamiltonian equation (usually consisting of coordinates and momenta) is not clear. The so-called Hamiltonian Q-learning takes minimization optimization and essentially claims an equivalence of energy in physics model, which might not always make sense. What is kinetic energy in your case? A Hamiltonian equation is usually defined as a functional of some parameters, is your problem defined over the policy?

2. In experiments, the proposed method is compared against Q-learning, but not with more advanced RL algorithms, e.g. DQN, DDPG, etc. The scalability of proposed method can be further investigated. More environments are also necessary.

3. It is not clear that how accurate it is to assume the Q-table to be a low-rank matrix. I agree it is a reasonable assumption, but highly doubt its practical performance, especially in large-scale problems.

4. The analysis in Section 3.3 is trivial, not enough to justify an acceptance.

Therefore, I am not convinced by the contributions.

---

> ### Author Response · Authors · 2020-11-22
> **Thanks so much for your feedback! We have added experimental results comparing our algorithm to advanced RL algorithms (DQN, Dueling DQN, and DDPG). We have also added theoretical results on sampling complexity.**
>
> Thank you for the constructive feedback! In the following, we address your concerns individually.
>
> --- **Ambiguity about equation (3)(9):** In fact, the Hamiltonian introduced in Section 2.2 is not related to the Q-value iteration over time steps governed by the Bellman equation (3)(9). The Hamiltonian dynamics is instead used for drawing samples from the distribution of future states for a given state-action pair. In this formulation, the "potential energy" is defined as the negative logarithm of the distribution of future states and the "kinetic energy" is defined as the quadratic function of an auxiliary variable which is drawn from a multivariate Gaussian distribution.
>
> --- **Experiments:** We have included additional experimental results in the revised version of our manuscript. We have carried out additional experiments to highlight the impact of using HMC sampling (instead of evaluating all possible next states) when DQN, Dueling DQN or DDPG is used for learning the optimal policy. Our results show that Hamiltonian $Q$-Learning needs a significantly smaller number of samples to converge to an $\epsilon$-optimal $Q$ function. In addition, we have also included results for the Acrobot stabilization task. Please see Section 4.1 and 4.2 for these changes.
>
> --- **Assumption on Low-rank Structure of the $Q$ matrix:** This assumption exploits the fact that the underlying state transition and reward function are endowed with some structure in a large class of problems. Prior work on value function approximation based approaches for reinforcement learning also leveraged a similar assumption implicitly while using various basis functions (e.g. CMAC, radial basis functions, diffusion wavelets etc.).  More recently, Yang et. al. ("Harnessing Structures for Value-Based Planning and Reinforcement Learning", ICLR 2020) have empirically shown that the $Q$-matrices for benchmark Atari games and classical control tasks exhibit low-rank structure. We have included a brief description of the rationale behind this assumption in Section 2.3 (Low-rank Structure in $Q$-learning and Matrix Completion) of our revised manuscript.
>
> --- **Analysis in Section 3.3:** In addition to our initial theoretical results on convergence guarantees under HMC sampling and boundedness guarantees under matrix completion with HMC sampling, we have incorporated an theoretical result (Theorem 3) on sampling complexity of our algorithm in the revised version.  We show that sampling complexity of our algorithm matches the lower bound of min-max sampling complexity. Also, to improve the flow of Section 3.3 and clarity of the contribution of this work, we have added brief descriptions to complement the mathematical details in Theorem 1, 2, and 3. Please see Section 3.3 and Appendix B for the related changes.

---

> > ### Comment · AnonReviewer2 · 2020-11-22
> > **Thanks for detailed revisions. I would increase my score by 1.**
> >
> > Thanks for detailed revisions, which addresses some of my concerns (NOT all). I would increase my score by 1.
> >
> > The major concern left is Hamiltonian equation as functionals, the physical soundness need more justification.

---

> > > ### Author Response · Authors · 2020-11-23
> > > **Thanks for your quick response!**
> > >
> > > We much appreciate your prompt response! In this work the Hamiltonian dynamics has been used only for drawing samples to estimate the expectation of $Q$ values associated with next states. The Hamiltonian dynamics is completely decoupled from the $Q$ function iteration over time.
> > >
> > > The Hamiltonian dynamics depend only on the state transition probability distribution - they have no dependency on the policy or the reward function. For a given transition probability kernel, we first define a target distribution function by multiplying the kernel with a smooth cut-off function (Please refer Eqn. 8). Then we define the “Potential Energy” of the Hamiltonian as
> > > \begin{equation} U(s)  =  -\log(\mathcal{P}(s)) = \frac{1}{2}(s-\mu)^T\Sigma^{-1}(s-\mu) - \frac{1}{2}\log\Big((2\pi)^{D_s}\det(\Sigma)\Big) - \sum_{i=1}^{D_s}\left[\log\Big(1+\exp(-\kappa (d^+_i-s^i))\Big) + \log\Big(1+\exp(-\kappa (s^i-d^-_i))\Big)\right]\end{equation}
> > >
> > > where, $\mu$ and $\Sigma$ correspond to the mean and variance of the transition probability kernel.  In the context of HMC $s$ is referred to as the position variable. The corresponding “Kinetic Energy” is then defined as
> > >
> > > \begin{equation} K(v)  =  -\log(\mathcal{P}(v|s))=\frac{1}{2}v^TM^{-1} v = \frac{1}{2}v^T\Sigma v. \end{equation}
> > >
> > > where $v$ is the momentum variable and $M=\Sigma^{-1}$ corresponds to the mass/inertia matrix associated with the Hamiltonian. As the Hamiltonian is the sum of the kinetic and the potential energy, i.e. $H(s,v) = U(s) + K(v)$, the Hamiltonian dynamics can be expressed as
> > >
> > > \begin{equation} \dot{s} = \frac{\partial K}{\partial v} = \Sigma v \end{equation}
> > >
> > > and
> > >
> > > \begin{equation} \dot{v} = - \frac{\partial U}{\partial s} = -\Sigma^{-1}(s -\mu) + \kappa\left[ S(-\kappa(d^+-s)) - S(-\kappa(s-d^-))\right], \end{equation}
> > >
> > > where $S(\xi_1,\cdots,\xi_{D_s}) = \big[S(\xi_1),\cdots,S(\xi_{D_s})\big]^T$ denotes element wise sigmoid function of the vector $\xi$.
> > >
> > > We initialize HMC sampling by drawing a random sample $s$ from the transition probability distribution and a new momentum variable $v$ from the multivariate Gaussian $ \mathcal{N}(0, \Sigma^{-1})$.
> > >
> > > We integrate the Hamiltonian dynamics for $L$ steps with step size $\Delta l$ to generate the trajectory from $(s,v)$ to $(s^{\prime},v^{\prime})$. To ensure that the Hamiltonian is conserved along the trajectory, we use a volume preserving symplectic integrator, in particular a leapfrog integrator which uses the following update rule to go from step $l$ to $l+\Delta l$:
> > >
> > > $\displaystyle v_{l+\frac{\Delta l}{2}}=v_l-0.5\Delta l \left.\frac{\partial U(s)}{\partial s} \right|_l,$
> > >
> > > $\displaystyle s_{l+\Delta l}=s_l+\Delta l \Sigma v_{l+\frac{\Delta l}{2}}$, and
> > >
> > > $\displaystyle v_{l+{\Delta l}}=v_l-0.5\Delta l \left.\frac{\partial U(s)}{\partial s} \right|_{l+\Delta l}.$
> > >
> > > Then, following the Metropolis–Hastings acceptance/rejection rule, this new proposal $(s^{\prime}, v^{\prime})$ is accepted with probability
> > >
> > > $\min\Big(1, \exp\left(H(s,v)-H(s^{\prime},-v^{\prime})\right)\Big)$.
> > >
> > > Let $\mathcal{H}_t$ be the set of next states obtained via HMC sampling, i.e., position variables from the accepted set of proposals. Then we update $Q(s_t,a_t)$ using Eqn. 9.
> > >
> > > We hope this answers your concern. If you think that this discussion would improve the clarity of our paper we can include in the next revision of our paper.
> > >
> > > **Update** We have added this discussion in Appendix E.

---

### Official Review · AnonReviewer1 · 2020-10-28
**Only applicable for tabular DP; experiments are weak**

**Rating:** 5
**Confidence:** 4

**Review:**

This work focuses on dynamic programming in the tabular setting. It proposes to use Hamiltonian Monte-Carlo (HMC) to sample the next states (instead of IID samples) and matrix completion to learn a low-rank Q matrix. It shows theoretical convergence. Experiments on discretized problems (CartPole and an ocean sampling problem) show that HMC and low-rank learning can behave more benignly compared to IID samples.

There are several issues with the current manuscript.

1. It can be applied to only tabular problems with known dynamics, which is restricted as many control problems are continuous, and their underlying dynamics are unknown. Q-learning can be applied in the RL setting, but not the method proposed in this paper.

2. Some technical details are not clearly described:
- What is the distribution used for Fig.1?
- Why is the \euler{P} in Eq.(5) necessary? Why not directly use \mathbb{P} as in Eq.(7) and (8)?

3. Experiments:
- It seems that the covariance matrices are defined arbitrarily for both CartPole and Ocean experiments.
- Do Fig.2(a) to (c) correspond to the same (optimal) actions? Or the optimal actions wrt their own Q values?
- Fig.2(d) is not very informative as we can have zero consecutive difference yet the Q function is far from the true optimal Q. Showing distance to the optimal Q values would be more meaningful.
- For the Ocean experiment, it is not clear how the action will affect the dynamic. More specifically, it is clear in Eq.(16) that the action $a$ will affect the state dynamic, but not for the ocean experiment equations on page 14.
- More descriptions about the ocean sampling problem would be helpful for the readers to understand the task. It seems that only the first term of the reward function on page 8 depends on the state (not sure where the action fits in this equation).

4. Paper organization. Both Sec.3.4 and Sec.4 are experiments, so why not put them into the same section?

Minors
- In Fig.1 caption, 1 should be 1/3.
- In reference, McAllister and Rasmussen: ta-efficient -> data-efficient

---

> ### Author Response · Authors · 2020-11-22
> **Thanks so much for your feedback! We have shown that Hamiltonian $Q$-Learning can be incorporated with DDPG, highlighting its scope beyond tabular problems.**
>
> Thank you for the constructive feedback! In the following, we address your concerns individually.
>
> --- **Applicable to only tabular problems with known dynamics:** Although we have used a finite MDP and a tabular $Q$ function, there is an underlying continuous dynamics. The discrete transition probability distribution that has been used for HMC sampling is also obtained by discretizing an underlying continuous transition distribution. Furthermore, our new experiments on sampling complexity highlights that Hamiltonian $Q$-Learning can be seamlessly incorporated with DDPG to improve its data efficiency (please refer Fig 3 and 4); it shows that our approach can be used for problems with continuous actions. In addition, we do not assume an explicit knowledge about the state transition dynamics; we only need access to a simulator that can generate next states from a given state-action pair.
>
> --- **Distribution used for Fig 1:** We used a Half normal distribution for the illustration in Fig 1.
>
> --- **Use of “\euler{p}” in Eqn 5:** In this work we assume that the finite MDP corresponds to an underlying continuous dynamics that have been discretized to define the finite MDP. Therefore we generate a discrete transition probability distribution from the corresponding continuous transition distribution.
>
> --- **Choice of Covariance Matrices for Numerical Experiments:** In our simulation results, we chose variance $\sigma$ in each dimension such that the range of the state space along the corresponding dimension is $3\sigma$ in each direction. Thus our results illustrate that Hamiltonain $Q$-Learning converges to $\epsilon$-optimal policies with significantly less number of samples even under high uncertainty in state transition. In our numerical experiments, uncertainty in state transition (covariance) corresponds to parameter variations, measurement noise and external disturbances. Thus we show that Hamiltonain $Q$-Learning archives desired control objective even under high variations, noise and disturbances.
>
> --- **Fig 2(a), 2(b) and 2(c):** Yes, all three figures correspond to the same (optimal) action.
>
> --- **Fig 2(d):** Figure 2(d) shows convergence of the $Q$ function. We agree with the reviewer that it does not show the accuracy of the converged $Q$ function. However, as explained in the previous comment since heat maps provided in figures (a) (c) have the same (optimal) action this shows the accuracy of the converged $Q$ function. As per reviewer’s suggestion we will replace figure (d) with a figure showing the error between $Q^t$ and optimal $Q^*$ in the 2nd revision that we are planning to upload within the next two days.
>
> ---  **For the ocean experiment it is not clear how the action affects the dynamic:** We apologize for the confusion. Action appears in the dynamic equations of the glider given in Appendix D. Last row of the matrix \tau contains the action $a$. We had accidentally written that as $u$ in the previous version. We have corrected that in the revised version. Please refer to the work of Rafael & Degani (“A Single-Actuated Swimming Robot: Design, Modelling, and Experiments”, Journal of Intelligent & Robotic Systems 2019) for more details.
>
> --- **More details would be helpful for ocean sampling example:** We apologize for not adding more details for the ocean sampling in this version. We will add it to the revised version we are planning to post within the next two days.
>
> --- **Paper organization in experimental results:** As per your suggestion we have included all the simulation results in Section 4 of this revised version.
>
> --- We have addressed your minor comments in our revised manuscript.

---

> > ### Comment · AnonReviewer1 · 2020-11-24
> > **Comments**
> >
> > Thank you for the additional comments and results. I have increased my score.
> >
> > Requiring a simulator for sampling is indeed weaker than knowing the whole dynamic model. However, it is still a strong requirement, so the applicability of the proposed method in real-world scenarios is limited.

---

> > > ### Author Response · Authors · 2020-11-25
> > > **Thank you for acknowledging our changes. Also, our approach can provide value even when a simulator is not available.**
> > >
> > > Thank you for your prompt response and for increasing our score. We agree that the use of simulators may be available for certain problems. However, simulators constitute a core component in an RL framework. Also, for problems without a simulator, our approach can still add value by reducing the amount of computations needed to process available data (which can be obtained from the past operational data of the real system) for computing $Q$ values.

---

> ### Author Response · Authors · 2020-11-23
> **Updated Figures 2(d) and 5(d); and Added details about the ocean sampling problem**
>
> We have a uploaded the second revision of our manuscript. In this revised version, Figure 2(d) and 5(d) show the distance between the optimal $Q^*$ and the $Q^t$ function learned via IID or HMC sampling. We have also included additional details about the ocean sampling problem/experiment in Section 4.3.

---

### Author Response · Authors · 2020-11-22
**Revised the paper to address reviewers' concerns**

Thank you for reviewing our paper! We much appreciate your insightful comments and constructive feedback. We have updated our paper to address your comments/concerns.  We have marked all the changes to the manuscript in blue.

Major revisions of the revised version are as follows:

--- **Experimental results:** We have significantly improved the experimental results section by adding the following additional results - (i) we have carried out additional experiments to highlight the impact of using HMC sampling (instead of evaluating all possible next states) when DQN, Dueling DQN or DDPG is used for learning the optimal policy; (ii) we have also included results for the Acrobot stabilization task. Please see Section 4.1 and 4.2 for these changes.

--- **Theoretical results:** We have significantly improved the theoretical contribution of the paper by adding a new theoretical result (Theorem 3 in Section 3.3 and its proof in Appendix B), which shows that the sampling complexity of our algorithm matches the mini-max sampling complexity lower bound.

---

### Decision · Program_Chairs · 2021-01-07
**Final Decision**

**Decision:**

Reject

**Comment:**


The paper considers exploiting low-rank structure in Q-function and the Hamiltonian Monte-Carlo (HMC) to approximate the expectation in Q-learning to reduce the stochastic approxiamtion error, and thus, achieves "efficient RL". The authors tested the algorithm empirically within some simple environments.

As reviewers (R1, R3, R4) mentioned, the major bottleneck of this algorithm is the assumption that the dynamics is known up to a constant, which is extremly strong, and thus, limits the application of the algorithm. I suggest the authors to consider the common RL setting, without any knowledge about the transition models, and make fair empirical comparison with baselines in the same setting.